# COMMUNICATION-EFFICIENT ALGORITHM FOR ASYNCHRONOUS MULTI-AGENT BANDITS

## ABSTRACT

We study the cooperative asynchronous multi-agent multi-armed bandits problem, where the active (arm pulling) decision rounds of each agent are asynchronous. In each round, only a subset of agents is active to pull arms, and this subset is unknown and time-varying. We propose a fully distributed algorithm that relies on novel asynchronous communication protocols. This algorithm attains near-optimal regret with constant (time-independent) communications for adversarial asynchronicity among agents. Furthermore, to protect the privacy of the learning process, we extend our algorithms to achieve local differential privacy with rigorous guarantees. Lastly, we report numerical simulations of our new asynchronous algorithms with other known baselines.

## 1 INTRODUCTION

Multi-Agent Multi-Armed Bandit (MA2B) is an important extension of the canonical multi-armed bandit model (Lai et al., 1985) in sequential decision making. A MA2B model consists of $K \in \mathbb{N}^+$ arms and $M \in \mathbb{N}^+$ agents. Each of these $K$ arms is associated with a reward distribution, and whenever an agent pulls an arm, they receive a reward sample drawn from their distribution. Most of MA2B prior works assume that agents are fully synchronous, to name a few (Bistritz & Bambos, 2020; Bistritz & Leshem, 2021; Boursier & Perchet, 2019; Chakraborty et al., 2017; Yang et al., 2023; Zhu et al., 2021; Wang et al., 2020b), where all $M$ agents have aligned decision rounds. That is, in each time slot (or interchangeably, "(decision) round"), each of these $M$ agents chooses one arm to pull and obtains a reward sample. The objective is to maximize the cumulative rewards of all agents over a given time horizon $T \in \mathbb{N}^+$ that is often large in practice.

In many real-world applications, instead of agents always being available in all time slots, they may be asynchronously available. For example, for clinical trial cooperation among multiple hospitals (i.e., agents) (Momtazmanesh et al., 2020; Aziz et al., 2021), whether a hospital can conduct a clinical treatment at a certain time depends on the availability and condition of the involved patients, which can be non-deterministic and heterogeneous. Another example is the path planning among multiple drones (i.e., agents) (Tsourdos et al., 2010). As each drone has multiple tasks to conduct, whether a drone is available for cooperative path planning at a certain time is also heterogeneous. Other examples of asynchronous agents include multiple secondary users sensing channel availability in cognitive radio network (Modi et al., 2017), multiple edge devices searching for efficient servers in edge computing (Dagan & Koby, 2018), multiple farms searching for right feed additives to cows for reducing methane emissions (Benetel et al., 2022), etc.

To model these asynchronous agent applications, in this paper, we propose and study the Asynchronous Multi-Agent Multi-Armed Bandits (AMA2B) model (§2). Instead of all agents being *active* (available) in each time slot, only a subset of agents is active for pulling arms. In other words, the active time slots of each agent can be unknown and irregularly spaced. One objective of AMA2B is to minimize regret, which is the aggregate cumulative differences between rewards from all active agents pulling the optimal arm and those generated by the cooperative algorithm's arm-pulling policy. Minimizing regret is equivalent to maximizing the total reward of all agents. As in a multi-agent system, the focus lies not solely on minimizing regret but also on developing efficient cooperative algorithms (i.e., with low communications) to achieve this goal.

Table 1: Communication cost comparison of `MA2B` algorithms in different scenarios

| `AMA2B` model | Algorithm | Communication |
|---|---|---|
| **Synchronous** | `DoE-bandit` | $O(KM \log \Delta_2^{-1})$ (Yang et al., 2023) |
| **Asynchronous** | `SE-ODC` | $O(KM^2 \Delta_2^{-2} \log T)$ (Chen et al., 2023) |
| **Asynchronous** | `SE-AAC-ODC` | $O(KM \log \Delta_2^{-1})$ (Theorem 2) |
| **$\varepsilon$-LDP Asynchronous** | `SE-AAC-ODC-CTB` | $O(KM \log(\Delta_2 \varepsilon)^{-1})$ (Theorem 9) |

$\Delta_2$ is the smallest reward gap.  All algorithms achieve near-optimal regrets.
For $\varepsilon$-LDP settings, we simplify factor $(e^\varepsilon + 1)/(e^\varepsilon - 1)$ by $O(1/\varepsilon)$.

In the synchronous `MA2B` literature, there are prior algorithms (Wang et al., 2020a; Yang et al., 2023) that achieve near-optimal regrets and *constant communication*[1]. However, these algorithms — relying on their aligned decision rounds — cannot be straightforwardly extended to `AMA2B`: The unaligned active decision rounds make it difficult for agents to efficiently cooperate (e.g., share reward observations). For example, instant broadcast of new observations to other agents, though a valid protocol, requires large $O(T)$ communications. On the other hand, aggregating multiple messages together into one communication may cause large delays because the sender agent may wait a large number of time slots to accumulate enough information due to a lack of active rounds. Meanwhile, during these time slots, other agents may have many active decision rounds and thus pay a high regret cost due to not being informed timely by the sender agent.

## 1.1 Contributions

First, in §3, we propose a constant communication algorithm for `AMA2B`; that is, the number of communications required is independent of time horizon $T$. The challenges for devising an efficient communication policy (for each agent) are (i) when to trigger communication and, if triggered, (ii) which agents to communicate with (§3.1). To tackle these challenges, we propose the Accuracy Adaptive Communication (`AAC`) protocol, which uses the relative amount of "new information" from recent local observations to decide when to communicate and the On-Demand Communication (`ODC`) protocol, which utilizes token exchanges to determine eligible agents to communicate with. Besides the communication protocols, we utilize the Success Elimination (`SE`) mechanism for arm pulling, which maintains a candidate arm set for pulling (initialized as the full arm set) and gradually eliminates suboptimal arms from this set until only one arm is left. We propose a cooperative bandit algorithm consisting of the combination of arm pulling policy `SE` with communication protocols `AAC` and `ODC`, named `SE-AAC-ODC` (§3.2). Then, in §4, we prove that `SE-AAC-ODC` achieves near-optimal regret while only requiring a constant number of communications, which is the first constant communication algorithm for `AMA2B`.

Second, we extend the algorithms to ensure privacy guarantees for the learning process. Toward this, we consider a local differential privacy (LDP) model, which protects the user's sensitive reward feedback from any attacker (e.g., any other user, an agent, or an adversary outside the agents). This privacy model is applicable to both fully distributed and leader-coordination schemes. We utilize a simple yet effective convert-to-Bernoulli mechanism to protect every possible observation. Through rigorous analysis, we show that the private algorithms achieve $\varepsilon$ user-level LDP (where $\varepsilon$ is the privacy budget). As a trade-off for our privacy protection, the regret and communication are increased by at most a factor $O((e^\varepsilon+1/e^\varepsilon-1)^2)$, which still maintains the constant communication. Due to the space limit, we defer the detail of the privacy results to Appendix A, which includes the formal privacy definitions (§A.1), privacy protection mechanisms (§A.2), theoretical analysis for our algorithm with privacy guarantee (§A.3), and numerical experiments (§A.4).

Finally, in §5, we conduct numerical simulations to evaluate the empirical performance of `SE-AAC-ODC` compared to other known baseline algorithms. The numerical results verify our theoretical results on achieving near-optimal group regret and constant communication costs.

---

[1]In this paper, we use the term "constant communication" to indicate that the number of communication is time-independent. Because in applications, the time horizon $T$ is usually much larger than the number of arms $K$ and the number of agents $M$.

## 1.2 RELATED WORKS

We compare the theoretical results of this paper and the most relevant prior results in Table 1, and in what follows we review them in detail.

Several prior works studied asynchronous multi-agent online learning models, e.g., in linear bandits (Li & Wang, 2022; He et al., 2022) and online convex optimization (Cesa-Bianchi et al., 2020; Jiang et al., 2021; Joulani et al., 2019). In the context of asynchronous multi-agent multi-armed bandits (AMA2B), the only prior work we know is Chen et al. (2023), where a fully distributed algorithm with an on-demand communication protocol (ODC) is proposed for AMA2B that achieves near-optimal regret at the expense of spending logarithmic communications. Although both Chen et al. (2023) and our SE-AAC-ODC algorithm utilize the ODC idea, it is not ODC itself that helps to reduce both algorithms' communications: Chen et al. (2023) leverage a buffer-threshold approach to achieve the communication cost of $O(KM^2\Delta_2^{-2}\log T)$. In contrast, SE-AAC-ODC introduces the AAC protocol to reduce the communication to $O(KM\log\Delta_2^{-1})$, where $\Delta_2$ is the smallest reward gap (formally defined in §2). We note that devising an algorithm combining AAC with ODC is different from their combination of the buffer-threshold approach and ODC, and the new communication protocol in this work is the first to achieve constant communications in AMA2B, which can be much smaller than the logarithmic one in Chen et al. (2023). Besides, Chen et al. (2023) do not consider differential privacy guarantee.

Among MA2B algorithms with constant communications (Wang et al., 2020b;a; Yang et al., 2023), Yang et al. (2023) is the most related work which proposed a constant communication algorithm with near-optimal problem-dependent regret performance for synchronous MA2B. Their algorithm relies on a distributed online estimator that only needs constant communications to guarantee that all agents' estimation is as good as a centralized estimation. However, the communication protocol in Yang et al. (2023) relies on the aligned decision rounds of all agents as their estimators require the knowledge of the total number of observations, while with the misaligned asynchronous decision rounds in AMA2B, obtaining such knowledge needs immediate communication in each time slot. To circumvent the misaligned decision round issue, we devise an asynchronous online estimator with a weaker guarantee — instead of all agents' estimators with good performance as in Yang et al. (2023), we show that at least one agent's estimator can have comparable performance to a centralized estimator. Then, we show that, with this weak estimator, we can devise a fully distributed algorithm that achieves the near-optimal regret with constant communications in AMA2B.

More broadly, synchronous MA2B has been extensively studied in either a fully distributed setting (Szorenyi et al., 2013; Chawla et al., 2020; Landgren et al., 2016; Buccapatnam et al., 2015; Martínez-Rubio et al., 2019; Bistritz & Bambos, 2020; Madhushani et al., 2021; Chakraborty et al., 2017; Cesa-Bianchi et al., 2016; Hillel et al., 2013; Dubey et al., 2020; Yang et al., 2021; 2022; Sankararaman et al., 2019; Féraud et al., 2019) or a leader-coordinated setting (Shi & Shen, 2021; Wang et al., 2020b;a; Bar-On & Mansour, 2019; Chakraborty et al., 2017; Dubey et al., 2020; Kolla et al., 2018), and various communication schemes such as peer-to-peer (Dubey & Pentland, 2020), consensus-based (Martínez-Rubio et al., 2019), gossip-style (Sankararaman et al., 2019; Chawla et al., 2020), and immediate broadcasting (Buccapatnam et al., 2015; Yang et al., 2021; 2022) have been considered. Beyond collaborative MA2B, there is also a branch of prior MA2B works studying a competitive setting where simultaneously pulling the same arm degrades the reward (Wang et al., 2020a; Boursier & Perchet, 2019; Shi et al., 2021; Bistritz & Leshem, 2018; Bubeck et al., 2020; Besson & Kaufmann, 2018). These works are at a clear distance from the asynchronous model studied in this paper.

## 2 MODEL

**Model** We consider an asynchronous multi-agent multi-armed bandits (AMA2B) model including $K$ arms and $M$ agents. Each arm $k \in \mathcal{K} := \{1, 2, \ldots, K\}$ is associated with a *Bernoulli* reward distribution with *unknown* mean $\mu_k$, and we assume $1 > \mu_1 > \mu_2 \geqslant \ldots \geqslant \mu_K > 0$ such that arm 1 is the unique optimal arm. We define $\Delta_k := \mu_1 - \mu_k$ for $k \geqslant 2$ as the reward gap between optimal arm 1 and suboptimal arm $k$. Each agent $m \in \mathcal{M} := \{1, 2, \ldots, M\}$ is associated with an

arbitrary sequence of activation times generated by an adversary at the beginning[2] and is unknown to agents. When an agent is activated, we refer to it as an *active* agent. Furthermore, we refer to the time slot that an agent becomes active as an (active) decision round because the agent needs to make a decision to pull an arm in the time slot.

Denote by $\mathcal{T}$ the set of time slots in which at least one agent is *active* and denote $T = |\mathcal{T}|$. In each time slot $t \in \mathcal{T} \coloneqq \{1, 2, \ldots, T\}$ ($T \in \mathbb{N}^+$), each active agent $m \in \mathcal{M}$ selects one arm $k \in \mathcal{K}$ to pull and obtains a reward $X_k^{(m)}(t)$ drawn from its reward distribution. We denote by $\mathcal{T}^{(m)}$ the set of time slots that agent $m$ is active, and $T^{(m)} \coloneqq |\mathcal{T}^{(m)}|$. We assume there are *no collisions*, i.e., when more than one agent pull the same arm, each of them gets a reward sample independently drawn from the reward distribution of this arm.

**Objective** We first define the expected regret of all agents as follows,

$$R(T) \coloneqq \sum_{m \in \mathcal{M}} \mathbb{E} \left[ T^{(m)} \mu_1 - \sum_{t \in \mathcal{T}^{(m)}} X_{k^{(m)}(t)}^{(m)} \right]$$

$$= \sum_{m \in \mathcal{M}} T^{(m)} \mu_1 - \mathbb{E} \left[ \sum_{m \in \mathcal{M}} \sum_{t \in \mathcal{T}^{(m)}} \mu_{k^{(m)}(t)} \right],$$

where the first expectation is taken over the randomness of the algorithm and rewards realization. In this cooperative AMA2B model, we allow agents to communicate with each other, and define the expected rounds of communications as follows,

$$C(T) \coloneqq \mathbb{E} \left[ \sum_{m \in \mathcal{M}} \sum_{t \in \mathcal{T}} \mathbb{1}\{\text{agent } m \text{ sends a message at time } t\} \right],$$

where $\mathbb{1}\{\cdot\}$ is the indicator function. Our objectives are to minimize both the group regret $R(T)$ and the number of communication rounds $C(T)$.

## 3 COMMUNICATION-EFFICIENT ALGORITHM FOR AMA2B

In this section, we introduce a fully distributed asynchronous algorithm, called SE-AAC-ODC (details summarized in §3.2), for the AMA2B model. SE-AAC-ODC involves multiple technical components (§3.1) — Accuracy Adaptive Communication (AAC), On-Demand Communication (ODC), and Successive Elimination (SE), which provide necessary functions to devise the algorithm. We also present the theoretical analysis of SE-AAC-ODC in §4. In the following, we first explain the technical challenges along with our algorithmic ideas for tackling the AMA2B model.

### 3.1 DESIGN CHALLENGES AND KEY IDEAS

The major challenge in designing a communication-efficient cooperation policy in synchronous MA2B is determining the right time for sharing local observations with other agents. By tackling this challenge using different techniques, recent works have significantly improved the communication cost of a bandit algorithm in the synchronous-agent setting (Yang et al., 2023; Wang et al., 2020a). However, in the asynchronous setting, and when the agents are not active in regular patterns, a new challenge is to find the right agent to communicate with. For example, in communications between two asynchronous agents — one "fast" agent (often active) and one "slow" agent (seldom active), a synchronous communication policy requires both agents to communicate in the same frequency, which would cause many redundant communications. Because, firstly, the slow agent may not have new observations to communicate with the fast agent, and, secondly, the slow agent, if there are no future active rounds, may not need the extra information from the fast agent. Putting together the existing challenge in synchronous MA2B and the new challenge in the asynchronous setting, a communication policy for AMA2B should appropriately answer two critical questions: (1) when to

---

[2]This kind of adversary is called *oblivious* in online learning because it cannot adaptively alter the active decision rounds based on history and algorithm's action.

---

**Algorithm 1** `SE-AAC-ODC`: successive arm elimination (for agent $m$)

---

1: **Inputs:** threshold parameter $\alpha > 1$
2: **Initialization:** $\tau_k(t) \leftarrow 0$, $\hat{\mu}_k^{(m)}(t)$, $n_k^{(m)}(t) \leftarrow 0$, $S_k^{(m)}(t) \leftarrow 0$, $\mathtt{tk}^{(m \to m')}$ for all agents $m' \in \mathcal{M}$ and arms $k \in \mathcal{K}$
3: **for all** $t \in \mathcal{T}$ **do**
4:      **if** $t \in \mathcal{T}^{(m)}$ **then**
5:          Update $\hat{\mu}_{k'}^{(m)}(t)$ for all arm $k' \in \mathcal{K}$ according to Eq. (3)
6:          Update the candidate arm set $\mathcal{C}(t)$ according to Eq. (4)          ▷ `Elimination`
7:          **if** any arm elimination happens **then** notify other agents for this elimination
8:          Pick an arm $k$ from candidate arm set $\mathcal{C}(t)$ to pull in a Round-Robin manner
9:          Obtain arm $k$'s reward observation $X_k^{(m)}(t)$
10:         $S_k^{(m)}(t) \leftarrow S_k^{(m)}(t) + X_k^{(m)}(t)$ and $n_k^{(m)}(t) \leftarrow n_k^{(m)}(t) + 1$
11:         **if** $\alpha \mathrm{ECR}_k^{(m)}(t) \leqslant \mathrm{ECR}_k^{(m)}(\tau_k(t))$ **then**     ▷ `AAC communication condition`
12:            $\tau_k(t) \leftarrow t$        ▷ `Update the latest communication time slot`
13:            Collect $(n_k^{(m')}(t), S_k^{(m')}(t))$ from all agents $m'$

                                     whose token $\mathtt{tk}^{(m \to m')}$ is held by agent $m$
14:            Update $n_k(t)$ and $\hat{\mu}_k(t)$
15:            Send $(k, n_k(t), \hat{\mu}_k(t), \tau_k(t))$ to other agents $m'$

                                     whose token $\mathtt{tk}^{(m \to m')}$ is held by agent $m$
16:         Send tokens $\mathtt{tk}^{(m \to m')}$ to corresponding agent $m'$
17:         Return all tokens $\mathtt{tk}^{(m' \to m)}$ on hold to corresponding agents $m'$
18:      **if** receive $\mathtt{tk}^{(m \to m')}$ from agent $m'$ **then**
19:         Send messages $(k', n_{k'}(t), \hat{\mu}_{k'}(t), \tau_{k'}(t))$ for arms whose updates were blocked

                               due to that agent $m$ did not hold token for agent $m'$
20:         Keep token $\mathtt{tk}^{(m \to m')}$
21:      **if** receive $\mathtt{tk}^{(m' \to m)}$ from agent $m'$ **then**
22:         Keep token $\mathtt{tk}^{(m' \to m)}$
23:      **if** receive broadcast $(k', n_{k'}(t), \hat{\mu}_{k'}(t), \tau_{k'}(t))$ from other agents $m'$ **then**
24:         Update local $(n_{k'}(t), \hat{\mu}_{k'}(t), \tau_{k'}(t))$ for arm $k'$

---

communicate, and (2) who to communicate with, both of which are challenging due to the agent's asynchronicity.

Our approach to addressing the question (1) is to have each agent wait until it has accumulated sufficient "new information" since its last communication. In other words, agents communicate when the expected "information gain" of the whole system is substantial enough. To measure the amount of new information, we use the ratio between the confidence radius based on an agent's local observations (including global observations obtained in the last communication) and the confidence radius only based on the global observation in the last communication. When this ratio exceeds a threshold, the "new information" contained in the agent's recent local observations is deemed worthwhile to share, and a new communication round is triggered.

Addressing question (2) regarding whom to communicate with, requires the determination of which agents can benefit from the new information. In other words, the new information should be sent to those agents whose regret may decrease as a consequence. The amount of decrease in agent regret depends on the agent's activation frequency, which can be addressed by introducing the idea of on-demand communication. That is, if an agent $m$ is on-demand (i.e., active), then other agents should share information with this agent $m$ to reduce its regret; otherwise, other agents shall stay away from communicating with inactive agents and save the cost of unnecessary communication.

In the following, we present how both high-level communication ideas can be implemented in a cooperative multi-agent bandit algorithm.

### 3.2 SE-AAC-ODC: A Fully Distributed Bandit Algorithm

#### 3.2.1 Accuracy Adaptive Communication (AAC) Protocol (Lines 11-15)

We use the confidence radius (half of a confidence interval's width) to represent the accuracy of the current estimate of reward mean and show how the AAC protocol determines when an agent decides to share information. We define the confidence radius as follows,

$$\text{CR}(n) := \min\{1, \sqrt{\frac{2 \log T}{n}}\}, \tag{1}$$

where $n$ is the number of samples (drawn from a Bernoulli distribution) used in this calculation. This confidence interval guarantees that, with a probability of at least $1 - T^{-4}$, the true reward mean $\mu$ lies inside the confidence interval $(\hat{\mu} - \text{CR}(n), \hat{\mu} + \text{CR}(n))$, where $\hat{\mu}$ is the empirical average of $n$ samples. To construct a confidence interval, the agents need to determine the number of observations $n$. However, since agents are distributed and asynchronous, without timely communication, each agent does not know the exact pull times for other agents since their last communication. To address this issue, we use the number of agent $m$'s recent local observations (since the last communication) as a surrogate for the number of other agents' recent local observations.

To facilitate presentation of the AAC protocol, we first fix an arm $k$ and consider the task that all asynchronous agents cooperate to estimate arm $k$'s mean $\mu_k$. Denote by $n_k^{(m)}(t)$ the number of observations of arm $k$ (excluding those received from others) by agent $m$ up to time slot $t$, and by $n_k(t)$ the total number of times among all agents that arm $k$ has been pulled on and before time slot $t$, i.e., $n_k(t) = \sum_{m \in \mathcal{M}} n_k^{(m)}(t)$. Denote the last communication rounds for sharing arm $k$'s observations at time slot $t$ as $\tau_k(t)$. We denote $\text{ECR}_k^{(m)}(t)$ an *estimated* confidence radius of agent $m$ for arm $k$ at time $t$ as a representation for accuracy, which can be expressed as

$$\text{ECR}_k^{(m)}(t) := \min\{1, \text{CR}(n_k(\tau_k(t)) + M(n_k^{(m)}(t) - n_k^{(m)}(\tau_k(t))))\}. \tag{2}$$

The 1 is in the min because arm reward means lie in $(0, 1)$ and the value 1 is the radius upper bound. The term $M(n_k^{(m)}(t) - n_k^{(m)}(\tau_k(t)))$ acts as a surrogate for the number of recent observations of other agents for arm $k$ since the last communication time $\tau_k(t)$ by using the agent $m$'s recent local observations. We note that if the current time slot is a communication round, i.e., $t = \tau_k(t)$, then there is no surrogate observation, therefore, the confidence radius is equal to the estimated one, i.e., $\text{CR}(n_k(\tau_k(t))) = \text{ECR}_k^{(m)}(\tau_k(t)), \forall m$. Hence, we refer to the estimated confidence radius in a communication round, i.e., $\text{ECR}_k^{(m)}(\tau_k(t))$ as the *aligned* confidence radius.

We use the ratio between the latest aligned confidence radius and the current estimated confidence radius $\text{ECR}_k^{(m)}(\tau_k(t))/\text{ECR}_k^{(m)}(t)$ to measure the relative amount of "new information" that agent $m$ collects since its last communication. Notice that the confidence radius decreases with the number of observations. Hence, when the ratio exceeds some predetermined threshold $\alpha > 1$, there is sufficient "new information" to initiate a new communication (Line 11).

#### 3.2.2 On-Demand Communication (ODC) Protocol (Lines 15-22)

We employ tokens to implement On-Demand Communication (ODC). At initialization, each agent $m$ is assigned $M-1$ tokens $\text{tk}^{(m \to m')}$, each corresponding to one agent $m' \in \mathcal{M} \setminus \{m\}$ (except for itself). Agent $m$ can communicate to agent $m'$ only when agent $m$ holds token $\text{tk}^{(m \to m')}$. More specifically, when the AAC communication condition is fulfilled (Line 11), if token $\text{tk}^{(m \to m')}$ is available at agent $m$, it sends a message containing the token $\text{tk}^{(m \to m')}$ to agent $m'$ (Lines 15-16); otherwise (without token $\text{tk}^{(m \to m')}$), agent $m$ does not communicate with agent $m'$. On the other hand, consider the case that agent $m$ holds some tokens $\text{tk}^{(m' \to m)}$ received from other agents $m'$. These tokens $\text{tk}^{(m' \to m)}$ are only useful for their corresponding agents $m'$ to decide whether to communicate with agent $m$ and are useless for agent $m$. Then, once agent $m$ is active he will immediately return these tokens to their corresponding senders $m'$, as a signal to agents $m'$ notifying that the agent $m$ is *on-demand* (Line 17). We note a special case: when agent $m'$ returns token $\text{tk}^{(m \to m')}$ to agent $m$, agent $m$ needs to send one updating message to agent $m'$ containing information that was not previously communicated due to agent $m$ not having a token for agent $m'$ (after this information update, agent $m$ keeps the returned token $\text{tk}^{(m \to m')}$) (Lines 18-20).

### 3.2.3 SUCCESSIVE ELIMINATION (SE) ARM PULL POLICY (LINES 6-10)

Denote by $\mathcal{C}(t)$ the candidate arm set, which is initialized as the full arm set, i.e., $\mathcal{C}(0) = \mathcal{K}$. The main idea of successive elimination is to uniformly explore all remaining arms in the candidate arm set in a round-robin manner (Line 8) and remove an arm from the candidate arm set (Line 6) whenever it is identified as suboptimal. Note that whenever an agent eliminates one arm, this agent notifies all other agents to eliminate the arm from their candidate arm sets as well; hence, all agents have the same candidate arm set (Line 7).

Next, we introduce notation to illustrate the technical details of eliminating a suboptimal arm from the candidate arm set. Denote by $S_k^{(m)}(t)$ the sum of $n_k^{(m)}(t)$ observations of arm $k$ for agent $m$ at time slot $t$, which can be expressed as

$$S_k^{(m)}(t) := \sum_{s=1}^{n_k^{(m)}(t)} X_k^{(m)}(s),$$

where $X_k^{(m)}(s)$ is a reward observation for arm $k$ of agent $m$. Next, we introduce estimator $\hat{\mu}_k^{(m)}(t)$ for the reward mean of agent $m$ for arm $k$ at time $t$ as follows,

$$\hat{\mu}_k^{(m)}(t) := \frac{n_k(\tau_k(t))\hat{\mu}_k(\tau_k(t)) + \left( S_k^{(m)}(t) - S_k^{(m)}(\tau_k(t)) \right)}{n_k(\tau_k(t)) + n_k^{(m)}(t) - n_k^{(m)}(\tau_k(t))}, \tag{3}$$

where $\tau_k(t)$ is the latest time slot (on or before $t$) that agent $m$ communicates information about arm $k$ to other agents (i.e., synchronizes globally), and $\hat{\mu}_k(\tau_k(t))$ is the average of all $n_k(\tau_k(t))$ observations. This estimator in Eq. (3) together with the confidence radius Eq. (1) yields the following confidence interval for $\mu_k$,

$$\mu_k \in \left( \hat{\mu}_k^{(m)} - \text{CR}\left( n_k(\tau_k(t)) + (n_k^{(m)}(t) - n_k^{(m)}(\tau_k(t))) \right), \right.$$
$$\left. \hat{\mu}_k^{(m)} + \text{CR}\left( n_k(\tau_k(t)) + (n_k^{(m)}(t) - n_k^{(m)}(\tau_k(t))) \right) \right).$$

With the above confidence interval, an arm $k$ is eliminated by agent $m$ from the candidate set $\mathcal{C}(t)$ at time $t$ if there exist an arm $k' \in \mathcal{C}(t)$ such that the upper confidence bound of arm $k$ is less than the lower confidence bound of arm $k'$, i.e.,

$$\hat{\mu}_k^{(m)}(t) + \text{CR}\left( n_k(\tau_k(t)) + (n_k^{(m)}(t) - n_k^{(m)}(\tau_k(t))) \right)$$
$$< \hat{\mu}_{k'}^{(m)}(t) - \text{CR}\left( n_{k'}(\tau_{k'}(t)) + (n_{k'}^{(m)}(t) - n_{k'}^{(m)}(\tau_{k'}(t))) \right). \tag{4}$$

Eq. (4) is the elimination condition used for identifying suboptimal arms in the candidate set.

## 4 THEORETICAL ANALYSIS OF SE-AAC-ODC

This section presents the theoretical analysis of the SE-AAC-ODC algorithm. We start by studying the estimation performance of the estimator in Eq. (3) in Lemma 1.

**Lemma 1.** *Assume $M$ agents independently and asynchronously sample arm $k$ associated with an i.i.d. reward process, Bernoulli distribution with unknown mean $\mu_k$, as Algorithm 1 (with threshold parameter $\alpha > 1$), and $n_k(t)$ is the total number of available samples across all agents. For any $t$, there exists an agent $\ell$ such that, with probability $1 - MT^{-3}$, we have $|\hat{\mu}_k^{(\ell)}(t) - \mu_k| \leqslant \alpha \text{CR}(n_k(t))$.*

Lemma 1 shows that among the estimates of all agents for reward mean $\mu_k$ of arm $k$, at least one agent $\ell$'s estimate $\hat{\mu}_k^{(\ell)}(t)$ enjoys the estimate accuracy comparable (up to an $\alpha$ factor) to that of an estimate that uses all of the observations of arm $k$, i.e., all $n_k(t)$ samples. In the proof of Lemma 1 deferred to Appendix B, this agent $\ell$ is set to be the one with the largest number of active decision rounds since the last communication time $\tau_k(t)$ for arm $k$. The active decision rounds of agents in AMA2B can vary in terms of the number of active rounds; hence, which agent is with a good estimator performance may change over time. Next, in Theorem 2, we show that although agents with good estimates can vary over time, the SE-AAC-ODC algorithm is able to adapt to the changes in the agent with the best estimates in the whole time horizon. Therefore, it achieves near-optimal regret and constant communication cost.

**Theorem 2.** *Algorithm 1's regret and communication can be upper bounded as*

$$R(T) \leqslant \sum_{k>1} \frac{8(1+\alpha)^2 \log T}{\Delta_k} + \sum_{k>1} M\Delta_k + KM^2, \tag{5}$$

$$C(T) \leqslant \sum_{k>1} 2M \log_\alpha \left(\frac{2(1+\alpha)}{\Delta_k}\right) + 2M \log_\alpha \left(\frac{2(1+\alpha)}{\Delta_2}\right) + 2KM^3. \tag{6}$$

To efficiently leverage the good estimators that may be of different agents in different time slots, SE-AAC-ODC employs an arm elimination notification technique (Line 7) such that whenever an agent is sufficiently confident to eliminate an arm, the agent notifies the other agents that this arm has been eliminated. We prove that the arm elimination notification technique ensures that agents lacking good estimates spend little time exploring such an arm in Lemma 12 (Appendix B), and this result guarantees the near-optimal regret upper bound in Theorem 2.

As Remark 4 shows, the major communication cost of SE-AAC-ODC comes from communications triggered by AAC. For any arm $k \in \mathcal{C}(t)$, AAC triggers one communication when the ratio $\text{ECR}_k^{(m)}(\tau_k(t))/\text{ECR}_k^{(m)}(t)$ is greater than the threshold $\alpha$ (Line 11), and AAC stops communicating about the arm $k$ when the arm is eliminated, which happens when $\text{CR}_k^{(m)}(t) < \Delta_k/2$, that is, $\text{ECR}_k^{(m)}(t) < c\Delta_k$ for some constant $c > 0$. Therefore, the total communication costs for this arm $k$ are upper bounded by $O(M \log_\alpha \Delta_k^{-1})$, where $M$ is because the communication is involved with all agents, the logarithm with base $\alpha$ is due to that the ratio based communication condition employs threshold $\alpha$, and the $\Delta_k^{-1}$ is because the communication stops when $\text{ECR}_k^{(m)}(t) < c\Delta_k$. We defer the detailed proof of Theorem 2 to Appendix B.

**Remark 3** (Regret optimality). *We recall the known MA2B's regret lower bound (Wang et al., 2020a, §1.2) (also proved for AMA2B by Chen et al. (2023, Appendix C)),*

$$\liminf_{T\to\infty} \frac{R(T)}{\log T} \geqslant \sum_{k>1} \frac{\Delta_k}{\text{KL}(\nu_k, \nu_1)}, \tag{7}$$

*where $\text{KL}(\cdot, \cdot)$ denotes the KL-divergence between two distributions, and $\nu_k$ denotes the reward distribution of arm $k$ with mean $\mu_k$. There are many distributions fulfilling that $\text{KL}(\nu_k, \nu_1) = \Theta(\Delta_k^2)$, e.g., Bernoulli or Gaussian, in which cases, the regret lower bound becomes*

$$\liminf_{T\to\infty} \frac{R(T)}{\log T} \geqslant C \cdot \sum_{k>1} \frac{1}{\Delta_k},$$

*where $C$ is a constant that depends on the specific reward distribution. On the other hand, the regret upper bound of SE-AAC-ODC in Eq. (5) can be rewritten as*

$$\limsup_{T\to\infty} \frac{R(T)}{\log T} \leqslant 8(1+\alpha)^2 \cdot \sum_{k>1} \frac{1}{\Delta_k}.$$

*Therefore, comparing the above asymptotic regret lower and upper bounds shows that the regret of the SE-AAC-ODC algorithm is tight up to a constant factor.*

**Remark 4** (Constant communication and its comparison). *There are three contributions to the communication cost of SE-AAC-ODC: (i) arm elimination notifications due to SE, (ii) token exchanges (send and return) for the implementation of ODC, and (iii) message sharing according to AAC. Communication cost due to (i) is $(K-1)(M-1)$ and does not appear in the communication upper bound in Eq. (6) as an independent term. Also, the communication cost due to (ii) is upper bounded by at most twice the communication cost of (iii). This is because token exchange consists of sending and returning: each token sending is always together with one message sending triggered by AAC (see Lines 15-16), and each token returning is just a consequence of the token's previous sending. Therefore, we only need to bound the communication costs of AAC, i.e., (iii), which yields the constant bound in Eq. (6). Lastly, we note that the only prior algorithm for AMA2B was proposed by Chen et al. (2023), which needed at least $O(KM^2\Delta_2^{-1} \log T)$ communications to achieve a near-optimal regret upper bound, while our SE-AAC-ODC only needs $O(KM \log \Delta_2^{-1})$ communications, which is much smaller than that of Chen et al. (2023) especially when $T$ is large.*

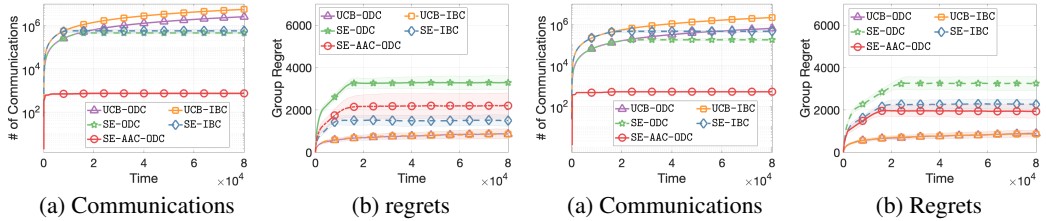

| (a) Communications | (b) regrets | (a) Communications | (b) Regrets |

| Figure 1: Experiment 1 (Stochastic) | Figure 2: Experiment 2 (Adversarial) |

Figures 1a and 2a show that `SE-AAC-ODC` enjoys a much smaller (constant) communication costs than known benchmark algorithms in both stochastic and adversarial `AMA2B`, and Figures 1b and 2b show that `SE-AAC-ODC` has similar near-optimal regret performance to known baselines.

**Remark 5.** *Recently, Yang et al. (2023) proposed a fully distributed cooperation algorithm for synchronous `MA2B` that achieved a near-optimal regret upper bound with constant communication cost. Their algorithm includes a confidence radius-based communication condition similar to Line 11 in Algorithm 1. However, Yang et al. (2023)'s algorithm design and analysis relied on the synchronous `MA2B` model (aligned decision rounds) and thus cannot be applied to `AMA2B`. Similar communication ideas were also applied to linear contextual bandits recently, e.g., Li & Wang (2022); He et al. (2022), but they still require logarithmic communications to achieve good regret performance.*

## 5 NUMERICAL EXPERIMENTS

Last, we conduct brief numerical experiments to verify the performance of the proposed algorithms using two experimental scenarios: stochastic and adversarial `AMA2B`.

*Experiment 1 (Stochastic).* In this experiment, we set $K = 16$ arms, 5 fast agents each with activation frequency 0.8, and 10 slow agents each with activation frequency 0.1. The arm reward means are drawn from the click-through-rates in the Ad-click dataset in Kaggle (2015). We report the number of communications and regrets after $T = 80\,000$ time slots averaged over 30 independent trials. We compare `SE-AAC-ODC` with $\alpha = 6$ against prior algorithms, including `UCB-ODC`, `UCB-IBC`, `SE-ODC` (AAE-ODC with constant buffer in Chen et al. (2023)), and `SE-IBC` (AAE-IBC in Chen et al. (2023)), with $\alpha = 3$ (the definition of their $\alpha$ parameter is different). The results, presented in Figure 1, show that the numbers of communications `SE-AAC-ODC` incur are significantly smaller than those incurred by benchmark algorithms, while the regret of `SE-AAC-ODC` is no worse than that of benchmark algorithms.

*Experiment 2 (Adversarial).* In this experiment, we study non-stationary asynchronicity. Specifically, we have $K = 16$ arms, $M = 10$ agents and the activation frequency of agent $m$ follows a sine function, $\sin(\theta_m + t/30)$, where the phase shifts $\theta_m = m/5$, $m \in \{1, ..., 10\}$ differ for different agents. We report the number of communications and regrets after $T = 80\,000$ time slots averaged over 30 independent trials. We compare `SE-AAC-ODC` with $\alpha = 5$ with prior algorithms, `UCB-ODC`, `UCB-IBC`, `SE-ODC`, and `SE-IBC`, with $\alpha = 3$. The results, presented in Figure 2, show that, in a non-stationary asynchronous setting, `SE-AAC-ODC` outperforms benchmark algorithms in terms of communication while obtain similar regrets.

## 6 FUTURE DIRECTIONS

This paper studied the asynchronous multi-agent multi-armed bandits problem and proposed a novel algorithms with near-optimal regret and horizon-independent communication costs for basic and private versions of the problem. An interesting future direction is, instead of user-level privacy, how to provide agent-level privacy protection while preserving near-optimal regret and constant communication costs. Besides the fully distributed algorithm studied in this paper, there is another kind of algorithm that can achieve constant communications (Wang et al., 2020a;b) in synchronous `MA2B` where they elect (or assume) a leader to coordinate the arm pulling among all agents. Design-

ing leader-coordinated algorithms that achieve constant communication in (asynchronous) `AMA2B` problem is another interesting future direction.

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

APPENDIX

## A   PRIVACY PROTECTION MECHANISMS

In multi-agent/distributed learning, it is critical to protect user's sensitive data from privacy risks, so as to encourage users to participate and collaborate with the agents in the learning process. Specifically, using the example mentioned in the introduction, imagine that $M$ hospitals (i.e., agents) are collaborating with each other to conduct a clinical trial (i.e., bandit problem) to study how different treatments (i.e., arms) can affect a disease. Each hospital will choose a specific treatment for participating patients (i.e., users) based on past observations of treatment effects. However, due to privacy concerns about health data leakage, the patients may not be willing to share the actual effects of the treatments with the hospital, which prevent the agents from learning from patient feedback. To ensure privacy, we use the notion of differential privacy (DP), which is a de facto standard for reasoning about information leakage (Dwork et al., 2014). As defined in Definition 6, DP implies that for any neighboring records, after an $\varepsilon$-DP mechanism, their statistical behaviors are indistinguishable. In this case, it is difficult for any attacker to determine which record is the source of the given output.

**Definition 6** ($\varepsilon$-differential privacy ($\varepsilon$-DP)). *For $\varepsilon > 0$, a randomized mechanism $Q : \mathcal{D} \to \mathbb{R}^l$ is said to be $\varepsilon$-DP on $\mathcal{D} \subset \mathbb{R}^{l'}$ if for any neighboring $x, x' \in \mathcal{D}$ where $\sum_{i \in [l']} \mathbb{1}\{x_i \neq x'_i\} = 1$ and a measurable subset $E$ of $\mathbb{R}^l$, we have $\mathbb{P}\{Q(x) \in E\} \leqslant e^\varepsilon \cdot \mathbb{P}\{Q(x') \in E\}$.*

In the above definition of DP, $\varepsilon$ is called the privacy budget, where smaller $\varepsilon$ implies higher levels of privacy protection. When $\varepsilon = \infty$, there is no privacy protection.

**Related works about differential privacy.**   There is extensive literature on studying differential privacy (DP) in multi-armed bandits (Tossou & Dimitrakakis, 2016; Basu et al., 2019; Shariff & Sheffet, 2018; Dubey & Pentland, 2020; Zhou & Chowdhury, 2023; Tenenbaum et al., 2021; Chowdhury & Zhou, 2022; Zhou & Chowdhury, 2023; Ren et al., 2020; Zheng et al., 2020; Li et al., 2022). Our work uses the notion of local DP, which has been studied under the single-agent MAB (Ren et al., 2020), the single-agent linear contextual MAB (Zheng et al., 2020), and distributed linear contextual bandits with partial feedback (Li et al., 2022). Ren et al. (2020) studies the DP in single-agent MAB, which largely inspires our privacy protection mechanism. However, we consider a new asynchronous multi-agent MAB setting and prove a series of important privacy/regret/communication guarantees. Other notions for DP, including central DP (Tossou & Dimitrakakis, 2016; Basu et al., 2019), shuffle DP (Tenenbaum et al., 2021; Chowdhury & Zhou, 2022; Zhou & Chowdhury, 2023), joint DP (Shariff & Sheffet, 2018; Dubey & Pentland, 2020; Zhou & Chowdhury, 2023), etc., pose weaker privacy protections for MAB models, but they may achieve better trade-offs between regret, communication, and privacy. Studying these DP notions for AMA2B will be left as interesting future works.

In this work, we focus on user-level local differential privacy (LDP), which allows the algorithm to be agnostic about privacy. For this reason, this notion is presently adapted by Apple and Google for their large-scale systems[3]. In what follows, we give its formal definitions (§A.1), mechanisms (§A.2), theoretical guarantees (§A.3), and numerical experiments (§A.4).

### A.1   USER-LEVEL LOCAL DIFFERENTIAL PRIVACY (LDP)

To define user-level LDP, we need to specify the user model and the threat model to supplement the model described in §2. Specifically, we denote $U = (u^{(m)}(t))_{t \in \mathcal{T}^{(m)}, m \in \mathcal{M}}$ be a sequence of $\sum_{m \in \mathcal{M}} T^{(m)}$ unique users. At each time $t \in \mathcal{T}^{(m)}$, user $u^{(m)}(t)$ comes to be served by agent $m$, who obtains reward $X_k^{(m)}(t)$ after the agent $m$ recommends arm $k^{(m)}(t)$ to the user. In this context, each user $u^{(m)}(t)$ is identified by their reward responses given to all possible actions recommended to them.

For the threat model, the users do not trust the agent or other users, and the privacy burden lies on the user himself. In this case, each user $u^{(m)}(t)$ needs to (1) release a private version of their reward

---

[3]https://desfontain.es/privacy/real-world-differential-privacy.html

---

**Algorithm 2** Convert-to-Bernoulli($\varepsilon$) (CTB ($\varepsilon$)) Mechanism

---

1: **Input:** A random reward $r \in [0, 1]$ from the user
2: **Return:** An independent sample following $Q(r) \sim \text{Bernoulli}(\frac{re^\varepsilon + 1 - r}{1 + e^\varepsilon})$

---

feedback $X_k^{(m)}(t)$ via a $\varepsilon$-DP mechanism $Q$, where $Q$ could be privacy protection software or trusted third-party plugins embedded in the user's devices or terminals, and the non-private data will not leave the control of the user unless they are processed and released by these software/plugins; and (2) the agents should make recommendations only based on the private releases. For any random vectors $X$ and $Y$, we use $X \in \sigma(Y)$ to denote that $X$ is determined by $Y$ plus some random factors independent of $Y$ and the bandit instance. Formally, user-level LDP is defined as follows in Definition 7.

**Definition 7** (User-level $\varepsilon$-LDP). *For $\epsilon > 0$, we say the learning process satisfies $\varepsilon$-LDP if (1) there is an $\varepsilon$-DP mechanism $Q : \mathcal{D} \to \mathbb{R}$, and (2) Action $k^{(m)}(t + 1) \in \sigma\big((k^{(j)}(s), Q(X_{k^{(j)}(s)}^{(j)}(s)))_{s \leqslant t, j \in \mathcal{M}}\big)$ for any agent $m$ and time $t$.*

Note that user-level LDP is a strong DP guarantee in the sense that it ensures that any attacker (which could be any other user, an agent, or an adversary outside the agents) cannot infer too much about any user's sensitive information (e.g., preference, reward feedback) or determine whether an individual participated in the learning process. Other DP notions, such as agent-level DP (where the agent containing the user can be trusted) (Zhou & Chowdhury, 2023), will also be satisfied by the post-processing of DP data if user-level LDP is preserved (Dwork et al., 2014).

## A.2 CONVERT-TO-BERNOULLI (CTB) MECHANISM

To guarantee user-level LDP, one can adopt the widely-used Laplace mechanism (Dwork et al., 2014) (i.e., adding Laplace noises to data records). However, adding Laplace noise makes the reward unbounded, which significantly increases regret and hinders the algorithm from obtaining constant communication costs. Inspired by Ren et al. (2020), we use a simple yet effective Convert-to-Bernoulli (CTB) mechanism, which converts the rewards bounded in $[0, 1]$ to Bernoulli responses. The CTB mechanism is described in Algorithm 2.

For our algorithm, CTB mechanism $Q$ is agnostic to the algorithm, in the sense that we only need to change the way the agent obtains the reward of Line 9 of Algorithm 1. In particular, agent $m$ now obtains arm $k$'s reward observation $Q(X_k^{(m)}(t))$ with parameter $\varepsilon$ as in Algorithm 2. The rest of the algorithm remains exactly the same.

## A.3 PRIVACY, REGRET, AND COMMUNICATION GUARANTEES

Here we present our privacy results, together with the new regret and communication cost. We also compare the non-private and private regret/communication cost.

**Theorem 8.** *SE-AAC-ODC (Algorithms 1) with CTB ($\varepsilon$) (Algorithm 2) all satisfy user-level $\varepsilon$-LDP.*

**Theorem 9.** *The regret and communication cost of Algorithm 1 with CTB ($\varepsilon$) (Algorithm 2) are upper bounded as follows,*

$$R(T) \leqslant \sum_{k>1} \frac{8(1+\alpha)^2 \log T}{\Delta_k} \cdot \left(\frac{e^\varepsilon + 1}{e^\varepsilon - 1}\right)^2 + \sum_{k>1} M\Delta_k + KM^2,$$

$$C(T) \leqslant \sum_{k>1} 2M \log_\alpha \left(\frac{2(1+\alpha)}{\Delta_k} \cdot \left(\frac{e^\varepsilon + 1}{e^\varepsilon - 1}\right)\right) + 2M \log_\alpha \left(\frac{2(1+\alpha)}{\Delta_2} \cdot \left(\frac{e^\varepsilon + 1}{e^\varepsilon - 1}\right)\right) + 2KM^3.$$

**Remark 10** (Regret comparison). *Compared with non-private SE-AAC-ODC in the fully distributed setting, the regret increases by at most a factor of $\left(\frac{e^\varepsilon+1}{e^\varepsilon-1}\right)^2$ factors, which is the cost for privacy protection. Since $\frac{e^\varepsilon+1}{e^\varepsilon-1} \leqslant 1 + \frac{2}{\varepsilon}$, this factor approaches 1 as $\varepsilon$ approaches infinity.*

**Remark 11** (Communication cost comparison). *When we compare communication cost bounds with non-private algorithms, private SE-AAC-ODC algorithm increases by at most a factor of $\log\left(\frac{e^\varepsilon+1}{e^\varepsilon-1}\right)$, which still remains independent of time horizon $T$.*

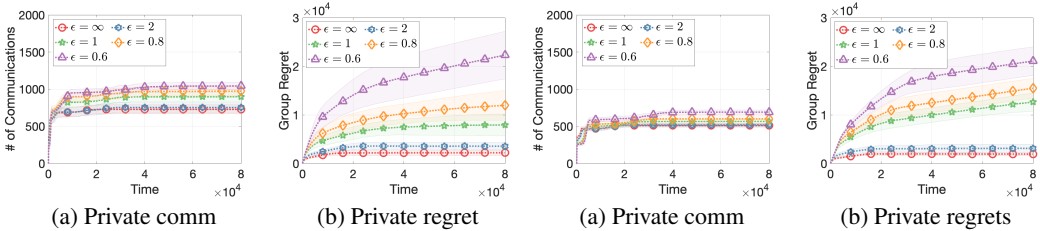

| (a) Private comm | (b) Private regret | (a) Private comm | (b) Private regrets |

Figure 3: Experiment 1 (Stochastic)          Figure 4: Experiment 2 (Adversarial)

Figures (c) (communications) and (d) (regrets) are performance of SE-AAC-ODC-CTB with privacy $\epsilon = \infty, 2, 1, 0.8, 0.6$.

### A.4 NUMERICAL EXPERIMENTS FOR PRIVACY PROTECTION MECHANISM

In this subsection, we report the empirical performance of SE-AAC-ODC-CTB algorithm with different privacy budgets $\epsilon = \infty, 2, 1, 0.8, 0.6$ in both stochastic and adversarial scenarios (see detail setup in §5). Figures 3a and 3b show that the number of communications and regrets of SE-AAC-ODC increase as the privacy level increases (i.e., as $\epsilon$ decreases) in stochastic AMA2B. Figures 4a and 4b show that the number of communications and regrets of SE-AAC-ODC increase as the privacy level $\epsilon$ increases in adversarial scenario as well.

## B PROOFS FOR FULLY DISTRIBUTED ALGORITHM

### B.1 PROOF OF LEMMA 1

We pick an agent $\ell$ with the highest number of times of pulling arm $k$ in the time slots from $\tau_k(t)$ to time $t$ (break tie arbitrarily), that is,

$$\ell := \arg\max_{m' \in \mathcal{M}} \sum_{s=\tau_k(t)}^{t} \mathbb{1}\{I^{(m')}(s) = k\} = \arg\max_{m' \in \mathcal{M}} \left(n_k^{(m')}(t) - n_k^{(m')}(\tau_k(t))\right), \qquad (8)$$

where $I^{(m')}(s)$ denotes the arm that agent $m'$ pulls at time slot $s$. Recall that the estimate $\hat{\mu}_k^{(\ell)}(t)$ is obtained by averaging $n_k(\tau_k(t)) + n_k^{(\ell)}(t) - n_k^{(\ell)}(\tau_k(t))$ samples, where $n_k(\tau_k(t))$ is the number of samples of arm $k$ among all agents at time slot $\tau_k(t)$. Hence, the following equation holds with probability $1 - MT^{-3}$,

$$
\begin{aligned}
|\hat{\mu}_k^{(\ell)}(t) - \mu_k| &\overset{(a)}{\leqslant} \mathrm{CR}(n_k(\tau_k(t)) + n_k^{(\ell)}(t) - n_k^{(\ell)}(\tau_k(t))) \\
&\overset{(b)}{\leqslant} \mathrm{CR}(n_k(\tau_k(t))) \\
&\overset{(c)}{<} \alpha \mathrm{CR}(n_k(\tau_k(t)) + M(n_k^{(\ell)}(t) - n_k^{(\ell)}(\tau_k(t)))) \\
&\overset{(d)}{<} \alpha \mathrm{CR}(n_k(t)),
\end{aligned}
$$

where inequality (a) is proved below by Hoeffding's inequality and union bound, inequality (b) is due to that the confidence radius becomes larger with a smaller number of samples, inequality (c) is due to that the condition in Line 11 of Algorithm 1 is false at time slot $t \, (> \tau_k(t))$, and inequality (d) is because that the agent $\ell$ has the highest number of times of pulling arm $k$ during $\tau_k(t)$ to $t$, that is, $n_k(\tau_k(t)) + M(n_k^{(\ell)}(t) - n_k^{(\ell)}(\tau_k(t))) \geqslant n_k(t)$.

Below, we present the detailed steps for proving inequality (a) as follows,

$$\mathbb{P}\left(|\hat{\mu}_k^{(\ell)}(t) - \mu_k| \leqslant \mathrm{CR}(n_k(\tau_k(t)) + n_k^{(\ell)}(t) - n_k^{(\ell)}(\tau_k(t)))\right)$$

$$= \mathbb{P}\left(|\hat{\mu}_k^{(\ell)}(t) - \mu_k| \leqslant \sqrt{\frac{4\log T}{n_k(\tau_k(t)) + n_k^{(\ell)}(t) - n_k^{(\ell)}(\tau_k(t))}}\right)$$

$$= 1 - \mathbb{P}\left(|\hat{\mu}_k^{(\ell)}(t) - \mu_k| > \sqrt{\frac{4\log T}{n_k(\tau_k(t)) + n_k^{(\ell)}(t) - n_k^{(\ell)}(\tau_k(t))}}\right)$$

$$\overset{(a1)}{\geqslant} 1 - \sum_{n=1}^{M \cdot t} \mathbb{P}\left(|\hat{\mu}_k^{(\ell)}(t) - \mu_k| > \sqrt{\frac{4\log T}{n}} \,\Big|\, n_k(\tau_k(t)) + n_k^{(\ell)}(t) - n_k^{(\ell)}(\tau_k(t)) = n\right)$$

$$\qquad\qquad \times \mathbb{P}\left(n_k(\tau_k(t)) + n_k^{(\ell)}(t) - n_k^{(\ell)}(\tau_k(t)) = n\right)$$

$$\geqslant 1 - \sum_{n=1}^{M \cdot t} \mathbb{P}\left(|\hat{\mu}_k^{(\ell)}(t) - \mu_k| > \sqrt{\frac{4\log T}{n}} \,\Big|\, n_k(\tau_k(t)) + n_k^{(\ell)}(t) - n_k^{(\ell)}(\tau_k(t)) = n\right)$$

$$\overset{(a2)}{\geqslant} 1 - \sum_{n=1}^{M \cdot t} T^{-4} \geqslant 1 - MT^{-3},$$

where inequality (a1) is due to union bound, and inequality (a2) is by applying Hoeffding's inequality.

## B.2 PROOF OF THEOREM 2

**Definitions.** Applying Hoeffding's inequality and union bound, we know that for any agent $m$, the following inequality holds with a probability of at least $1 - MT^{-3}$,

$$|\hat{\mu}_k^{(m)}(t) - \mu_k| \leqslant \mathrm{CR}(n_k(\tau_k(t)) + (n_k^{(m)}(t) - n_k^{(m)}(\tau_k(t)))), \tag{9}$$

where we recall that $\tau_k(t)$ is the last time slot that agents conduct communication to synchronize the observations of arm $k$ before time slot $t$.

We denote the decision made at a time slot $t$ as a Type-I decision when Eq. (9) holds for any agent $m \in \mathcal{M}$ and arm $k \in \mathcal{K}$ at this time slot $t$; otherwise, we denote it as a Type-II decision.

We note that, as long as the decision is Type-I, the elimination condition in Eq. (4) always correctly eliminates suboptimal arms, and, therefore, when there is only one arm remaining in the candidate arm set $\mathcal{C}(t)$, it is the optimal arm for sure. In the next two steps, we bound the probability that there are any Type-II decisions and the number of pulling times of any suboptimal arm when there are only Type-I decisions respectively.

**Step 1. Upper bound the probability of any Type-II decision occurring.** Below we bound the probability of the event that there exists any confidence interval not containing its corresponding true reward mean.

$$\mathbb{P}(\exists(k, m, t), |\hat{\mu}_k^{(m)}(t) - \mu_k| > \mathrm{CR}(n_{k'}(\tau_k(t)) + (n_k^{(m)}(t) - n_k^{(m)}(\tau_{k'}(t)))))$$

$$\leqslant \mathbb{P}(\exists(k, m, t, n), |\hat{\mu}_k^{(m)}(t) - \mu_k| > \mathrm{CR}(n))$$

$$\leqslant \sum_{(k,m,t,n) \in (\mathcal{K} \times \mathcal{M} \times \mathcal{T} \times \mathcal{T})} \mathbb{P}(|\hat{\mu}_k^{(m)}(t) - \mu_k| > \mathrm{CR}(n))$$

$$\leqslant \sum_{(k,m,t,n) \in (\mathcal{K} \times \mathcal{M} \times \mathcal{T} \times \mathcal{T})} MT^{-3} = KM^2 T^{-1}.$$

**Step 2. Upper bound the number of times of pulling suboptimal arms.**

**Lemma 12.** *At any time $t \leqslant T$, if the optimal arm lies in the candidate set and an agent $m$ makes a Type-I decision with pulling a suboptimal arm $k$, i.e., $I^{(m)}(t) = k$, we have $n_k(t) \leqslant$*

$8(1+\alpha)^2 \log T / \Delta_k^2$. *Therefore, the total number of pulling times of arm $i$ in the whole time horizon is upper bounded as follows,*

$$n_k(T) \leqslant \frac{8(1+\alpha)^2 \log T}{\Delta_k^2} + M.$$

*Proof.* Arm $k$ is pulled at some time $t$

$$\overset{(a)}{\implies} k \in \mathcal{C}(t) \text{ for agent } \ell \text{ fulfills Lemma 1}$$

$$\overset{(b)}{\implies} \hat{\mu}_k^{(\ell)}(t) + \text{CR}(n_k(\tau_k(t)) + M(n_k^{(\ell)}(t) - n_k^{(\ell)}(\tau_k(t))))$$

$$\geqslant \hat{\mu}_{k'}^{(\ell)}(t) - \text{CR}(n_{k'}(\tau_{k'}(t)) + M(n_{k'}^{(\ell)}(t) - n_{k'}^{(\ell)}(\tau_{k'}(t)))) \text{ for any } k' \in \mathcal{C}(t)$$

$$\overset{(c)}{\Longleftrightarrow} \hat{\mu}_k^{(\ell)}(t) + 2\text{CR}(n_k(\tau_k(t)) + M(n_k^{(\ell)}(t) - n_k^{(\ell)}(\tau_k(t)))) \geqslant \hat{\mu}_{k'}^{(\ell)}(t) \text{ for any } k' \in \mathcal{C}(t)$$

$$\overset{(d)}{\implies} \hat{\mu}_k^{(\ell)}(t) + 2\text{CR}(n_k(t)) \geqslant \hat{\mu}_{k'}^{(\ell)}(t) \text{ for any } k' \in \mathcal{C}(t)$$

$$\implies \hat{\mu}_k^{(\ell)}(t) + 2\text{CR}(n_k(t)) \geqslant \hat{\mu}_1^{(\ell)}(t)$$

$$\overset{(e)}{\implies} \mu_k + (2+\alpha)\text{CR}(n_k(t)) \geqslant \mu_1 - \alpha\text{CR}(n_1(t))$$

$$\implies 2(1+\alpha)\text{CR}(n_k(t)) \geqslant \mu_1 - \mu_k = \Delta_k \tag{10}$$

$$\implies n_k(t) \leqslant \frac{8(1+\alpha)^2 \log T}{\Delta_k^2},$$

where (a) is because the candidate arm sets $\mathcal{C}(t)$ are the same for all agents (including arm $\ell$), (b) is by the definition of candidate arm set, (c) is because arms in the candidate arm set are evenly explored in a round-robin manner, (d) is from the definition of arm $\ell$ in Eq. (8), and (e) is by applying Lemma 1.

Lastly, since the pulling of arm $k$ in the critical time slot $t$ is not counted, the total pulling times of arm $k$ may be increased by $M$ at most, i.e.,

$$n_k(T) \leqslant \frac{8(1+\alpha)^2 \log T}{\Delta_k^2} + M.$$

$\square$

Combining the results of Steps 1 and 2, the group regret is upper bounded as follows,

$$\mathbb{E}[R(T)] \leqslant \sum_{k>1} n_k(T) \times \Delta_k + KM^2 T^{-1} \times T$$

$$\leqslant \sum_{k>1} \frac{8(1+\alpha)^2 \log T}{\Delta_k} + \sum_{k>1} M\Delta_k + KM^2$$

**Step 3. Upper bound communication costs** If there are any Type-II decisions, the total communication times is at most $KM^2$.

Assume there is no Type-II decision. In the proof of Lemma 12, we have a middle step Eq. (10): for any suboptimal arm $k$, denoting $\kappa_k$ as the last time slot that the arm was pulled, we have

$$2(1+\alpha)\text{CR}(n_k(\kappa_k)) \geqslant \Delta_k.$$

Recall that $\tau_k(t)$ is the latest communication round about arm $k$ on or before time slot $t$. Then, $\text{ECR}_k^{(m)}(\tau_k(\kappa_k))$ is the ECR at the latest communication for arm $k$ which can be upper bounded as follows,

$$\text{ECR}_k^{(m)}(\tau_k(\kappa_k)) \overset{(a)}{=} \text{CR}(n_k(\tau_k(\kappa_k))) \geqslant \text{CR}(n_k(\kappa_k)) \geqslant \frac{\Delta_k}{2(1+\alpha)},$$

where equality (a) is because $\tau_k(\kappa_k)$ is a communication time slot in which the estimated confidence radius (ECR) is equal to CR. Recall the initial $\text{ECR}_k^{(m)}(0) = 1$ by definition. The total number of

times of communication on arm $k$ is upper bounded as follows,

$$\log_\alpha \left( \frac{\text{ECR}_k^{(m)}(0)}{\text{ECR}_k^{(m)}(\tau_k(\kappa_k))} \right) \leqslant \log_\alpha \left( \frac{2(1+\alpha)}{\Delta_k} \right).$$

Since all arms are pulled in a round-robin manner, the communication cost on the optimal arm is upper bounded by $\log_\alpha \left( \frac{2(1+\alpha)}{\Delta_2} \right)$ where $\Delta_2$ is the smallest reward gap.

Summing the above two type cases yields the communication upper bound as follows,

$$\sum_{k>1} \log_\alpha \left( \frac{2(1+\alpha)}{\Delta_k} \right) + \log_\alpha \left( \frac{2(1+\alpha)}{\Delta_2} \right) + KM^2.$$

As each communication round above needs $2(M-1)$ communications and the notification of arm elimination needs $(K-1)M$ communications in total, the final communication costs are upper bounded by

$$\sum_{k>1} 2M \log_\alpha \left( \frac{2(1+\alpha)}{\Delta_k} \right) + 2M \log_\alpha \left( \frac{2(1+\alpha)}{\Delta_2} \right) + 2KM^3.$$

## C   PROOFS FOR PRIVACY PROTECTION MECHANISM

### C.1   PROOF OF THEOREM 8

*Proof of Theorem 8.* To prove the privacy guarantee, we need to check the two requirements of the Definition 7.

For requirement i), we rely on the key proposition as follows, which proves that CTB $(\varepsilon)$ is $\varepsilon$-DP.

**Proposition 13** (Lemma 5 of Ren et al. (2020)). *CTB $(\varepsilon)$ mechanism (Algorithm 2) is $\varepsilon$-DP on [0,1], and the returned sample follows the Bernoulli distribution with mean $\mu_{k,\varepsilon} = 1/2 + (\mu_k - 1/2) \cdot \frac{e^\varepsilon - 1}{e^\varepsilon + 1}$.*

For requirement ii), since the private version of SE-AAC-ODC-CTB only use the observation from CTB, which satisfies requirement ii). □

### C.2   PROOF OF THEOREM 9

*Proof of Theorem 9.* Notice that after applying CTB, Proposition 13 ensures that the means of the private (Bernoulli) observations become $1 > \mu_{1,\varepsilon} > \mu_{2,\varepsilon} \geqslant \ldots \geqslant \mu_{K,\varepsilon} > 0$, where arm 1 is still the unique optimal arm. For the sub-optimal arm $k$, the sub-optimality gap becomes $\Delta_{k,\varepsilon} \triangleq \frac{e^\varepsilon - 1}{e^\varepsilon + 1} \Delta_k$ when bounding the number of times $n_k(T)$ until sub-optimal arms will not be pulled. For each pull of arm $k$, the algorithm will still pay $\Delta_k$ regret. Specifically, we slightly adapt the proof from Section B and show the following lemma.

**Lemma 14.** *Assume $M$ agents independently sample an arm $k$ associated with an i.i.d. reward process with unknown mean $\mu_k$ as Algorithm 1 (with threshold parameter $\alpha > 1$), and $n_k(t)$ is the total available samples of all agents. For any $t$, there exists an agent $\ell$ such that, with probability $1 - MT^{-3}$, we have*

$$|\hat{\mu}_k^{(\ell)}(t) - \mu_{k,\varepsilon}| \leqslant \alpha \text{CR}(n_k(t)).$$

*Proof.* We pick an agent $\ell$ with the highest number of times of pulling arm $i$ in the time slots from $\tau_k(t)$ to time $t$, that is,

$$\ell \in \arg\max_{m' \in \mathcal{M}} \sum_{s=\tau_k(t)}^{t} \mathbb{1}\{I^{(m')}(s) = i\}. \tag{11}$$

Note that the estimate $\hat{\mu}_k^{(\ell)}(t)$ is obtained by averaging $n_k(\tau_k(t)) + n_k^{(\ell)}(t) - n_k^{(\ell)}(\tau_k(t))$ samples. Hence, the following equation holds with probability $1 - MT^{-3}$.

$$
\begin{aligned}
|\hat{\mu}_k^{(\ell)}(t) - \mu_{k,\varepsilon}| &\overset{(a)}{\leqslant} \mathrm{CR}(n_k(\tau_k(t)) + n_k^{(\ell)}(t) - n_k^{(\ell)}(\tau_k(t))) \\
&\overset{(b)}{\leqslant} \mathrm{CR}(n_k(\tau_k(t))) \\
&\overset{(c)}{<} \alpha\mathrm{CR}(n_k(\tau_k(t)) + M(n_k^{(\ell)}(t) - n_k^{(\ell)}(\tau_k(t)))) \\
&\overset{(d)}{<} \alpha\mathrm{CR}(n_k(t)),
\end{aligned}
$$

where inequality (a) is by Hoeffding's inequality and union bound, inequality (b) is due to that the confidence radius becomes larger with a smaller number of samples, inequality (c) is due to that the condition in Line 11 is false at time slot $t\,(> \tau_k(t))$, and inequality (d) is because that the agent $\ell$ has the highest number of times of pulling arm $k$ during $\tau_k^{(\ell)}(t)$ to $t$, that is, $n_k(\tau_k(t)) + M(n_k^{(\ell)}(t) - n_k^{(\ell)}(\tau_k(t))) \geqslant n_k(t)$. $\qquad\square$

Now we follow steps 1-3 in Appendix B.

**Step 1. Upper bound the probability of any Type-II decision occurring.**

$$
\begin{aligned}
&\mathbb{P}(\exists(k,m,t), |\hat{\mu}_k^{(m)}(t) - \mu_{k,\varepsilon}| > \mathrm{CR}(n_{k'}(\tau_k(t)) + (n_k^{(m)}(t) - n_k^{(m)}(\tau_{k'}(t))))) \\
&\leqslant \mathbb{P}(\exists(k,m,t,n), |\hat{\mu}_k^{(m)}(t) - \mu_{k,\varepsilon}| > \mathrm{CR}(n)) \\
&\leqslant \sum_{(k,m,t,n)\in(\mathcal{K}\times\mathcal{M}\times\mathcal{T}\times\mathcal{T})} \mathbb{P}(|\hat{\mu}_k^{(m)}(t) - \mu_{k,\varepsilon}| > \mathrm{CR}(n)) \\
&\leqslant \sum_{(k,m,t,n)\in(\mathcal{K}\times\mathcal{M}\times\mathcal{T}\times\mathcal{T})} MT^{-3} = KM^2 T^{-1}.
\end{aligned}
$$

**Step 2. Upper bound the number of times of pulling suboptimal arms**

**Lemma 15.** *At any time $t \leqslant T$, if the optimal arm lies in the candidate set and an agent makes a Type-I decision with pulling a suboptimal arm $i$, i.e., $I^{(m)}(t) = i$, we have $n_k(t) \leqslant 8(1+\alpha)^2 \log T/\Delta_{k,\varepsilon}^2$. Therefore, the total number of pulling times of arm $i$ in the whole time horizon is upper bounded as follows,*

$$
n_k(T) \leqslant \frac{8(1+\alpha)^2 \log T}{\Delta_{k,\varepsilon}^2} + M.
$$

Combining the results of Steps 1 and 2, the group regret is upper bounded as follows,

$$
\begin{aligned}
\mathbb{E}[R(T)] &\leqslant \sum_{k>1} n_k(T) \times \Delta_k + KM^2 T^{-1} \times T \\
&\leqslant \sum_{k>1} \frac{8(1+\alpha)^2 \log T}{\Delta_k} \cdot \left(\frac{e^\varepsilon + 1}{e^\varepsilon - 1}\right)^2 + \sum_{k>1} M\Delta_k + KM^2.
\end{aligned}
$$

**Step 3. Upper bound communication costs** For communication, the bound also holds by replacing $\Delta_k$ with $\Delta_{k,\varepsilon}$. Specifically, if there are any Type-II decisions, the total communication times is at most $KM^2$.

Assume there is no Type-II decision. Following the proof of Lemma 12, we have a similar middle step Eq. (12): for any suboptimal arm $k$, denoting $\kappa_k$ as the last time slot that the arm was pulled, we have

$$
2(1+\alpha)\mathrm{CR}(n_k(\kappa_k)) \geqslant \Delta_{k,\varepsilon}.
$$

Hence, the last $\mathrm{ECR}_k^{(m)}(T)$ can be upper bounded as

$$
\mathrm{ECR}_k^{(m)}(T) \overset{(a)}{=} \mathrm{CR}(n_k(\kappa_k)) \geqslant \frac{\Delta_{k,\varepsilon}}{2(1+\alpha)},
$$

where inequality (a) is because after round $\kappa_k$ there is no further pulling on arm $k$. Recall the initial $\text{ECR}_k^{(m)}(0) = 1$. The total number of times of communication on arm $i$ is upper bounded as follows,

$$\log_\alpha \left( \frac{\text{ECR}_k^{(m)}(0)}{\text{ECR}_k^{(m)}(T)} \right) \leqslant \log_\alpha \left( \frac{2(1+\alpha)}{\Delta_{k,\varepsilon}} \right).$$

Since all arms are pulled in a round-robin manner, the communication cost on the optimal arm is upper bounded by $\log_\alpha \left( \frac{2(1+\alpha)}{\Delta_{2,\varepsilon}} \right)$ where $\Delta_{2,\varepsilon}$ is the smallest reward gap.

Summing the above two type cases yields the communication upper bound as follows,

$$\sum_{k>1} \log_\alpha \left( \frac{2(1+\alpha)}{\Delta_k} \cdot \left( \frac{e^\varepsilon + 1}{e^\varepsilon - 1} \right) \right) + \log_\alpha \left( \frac{2(1+\alpha)}{\Delta_2} \cdot \left( \frac{e^\varepsilon + 1}{e^\varepsilon - 1} \right) \right) + KM^2.$$

As each communication round above needs $2(M-1)$ communications and the notification of arm elimination needs $(K-1)M$ communications in total, the final communication costs are upper bounded by

$$\sum_{k>1} 2M \log_\alpha \left( \frac{2(1+\alpha)}{\Delta_k} \cdot \left( \frac{e^\varepsilon + 1}{e^\varepsilon - 1} \right) \right) + 2M \log_\alpha \left( \frac{2(1+\alpha)}{\Delta_2} \cdot \left( \frac{e^\varepsilon + 1}{e^\varepsilon - 1} \right) \right) + 2KM^3.$$

$\square$

*Proof of Lemma 15.*

Arm $k$ is pulled at time $t$

$\stackrel{(a)}{\Longrightarrow} k \in \mathcal{C}(t)$ for agent $\ell$ fulfills Lemma 14

$\stackrel{(b)}{\Longrightarrow} \hat{\mu}_k^{(\ell)}(t) + \text{CR}(n_k(\tau_k(t)) + M(n_k^{(\ell)}(t) - n_k^{(\ell)}(\tau_k(t))))$

$\qquad \geqslant \hat{\mu}_{k'}^{(\ell)}(t) - \text{CR}(n_{k'}(\tau_{k'}(t)) + M(n_{k'}^{(\ell)}(t) - n_{k'}^{(\ell)}(\tau_{k'}(t))))$ for any $k' \in \mathcal{C}(t)$

$\stackrel{(c)}{\Longrightarrow} \hat{\mu}_k^{(\ell)}(t) + 2\text{CR}(n_k(\tau_k(t)) + M(n_k^{(\ell)}(t) - n_k^{(\ell)}(\tau_k(t)))) \geqslant \hat{\mu}_{k'}^{(\ell)}(t)$ for any $k' \in \mathcal{C}(t)$

$\stackrel{(d)}{\Longrightarrow} \hat{\mu}_k^{(\ell)}(t) + 2\text{CR}(n_k(t)) \geqslant \hat{\mu}_{k'}^{(\ell)}(t)$ for any $k' \in \mathcal{C}(t)$

$\Longrightarrow \hat{\mu}_k^{(\ell)}(t) + 2\text{CR}(n_k(t)) \geqslant \hat{\mu}_1^{(\ell)}(t)$

$\stackrel{(e)}{\Longrightarrow} \mu_{k,\varepsilon} + (2+\alpha)\text{CR}(n_k(t)) \geqslant \mu_{1,\varepsilon} - \alpha\text{CR}(n_1(t))$

$\Longrightarrow 2(1+\alpha)\text{CR}(n_k(t)) \geqslant \mu_{1,\varepsilon} - \mu_{k,\varepsilon} = \Delta_{k,\varepsilon}$ $\qquad$ (12)

$\Longrightarrow n_k(t) \leqslant \frac{8(1+\alpha)^2 \log T}{\Delta_{k,\varepsilon}^2}$

where (a) is because the candidate arm sets $\mathcal{C}(t)$ are the same for all agents (including arm $\ell$), (b) is by the definition of candidate arm set, (c) is because arms in the candidate arm set are evenly explored in a round-robin manner, (d) is from the definition of arm $\ell$ in Eq. (11), and (e) is by applying Lemma 14.

Lastly, since the pulling of arm $i$ in the critical time slot $t$ is not counted, the total pulling times of arm $i$ may be increased by $M$ at most, i.e.,

$$n_k(T) \leqslant \frac{8(1+\alpha)^2 \log T}{\Delta_{k,\varepsilon}^2} + M.$$

$\square$

# D  ADDITIONAL SIMULATIONS

In this section, we report three more experiments to further corroborate the performance of our algorithm and compare it to baselines.

In Figure 5, we report the `SE-ODC` with no buffer (green), and `SE-ODC` with doubling buffer (brown) and compare both to our algorithm, `SE-AAC-ODC` (red). Although `SE-ODC-D` with doubling buffer enjoys better communication performance than `SE-ODC` with constant buffer, `SE-ODC-D`'s communication performance is still worse than that of `SE-AAC-ODC` proposed in our paper, and `SE-ODC-D`'s empirical regret is far worse than `SE-AAC-ODC`. We updated this additional simulation results in the revised version of this paper.

Figure 6 compares the performance of `SE-AAC-ODC` with $\alpha = 3$ and $\alpha = 6$. This simulation validates the communication-regret trade-off—the case of $\alpha = 3$ enjoys a better regret but suffers a higher communication. Additional, the communications of both cases are much lower than that of other baseline algorithms.

In Figure 7, we report additional experiments where the number of agents are $M = 30$ greater than the number of arms $K = 16$. In these experiments, our algorithm outperforms other baseline algorithms in communication and enjoys similar regret as the other baselines, which are the same observation as in our original numerical simulations at Figure 1.

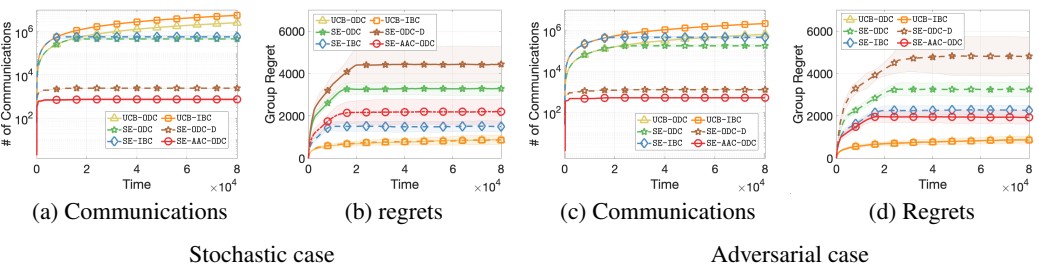

(a) Communications     (b) regrets     (c) Communications     (d) Regrets

Stochastic case        Adversarial case

Figure 5: Additional experiments for `SE-ODC` with doubling buffer

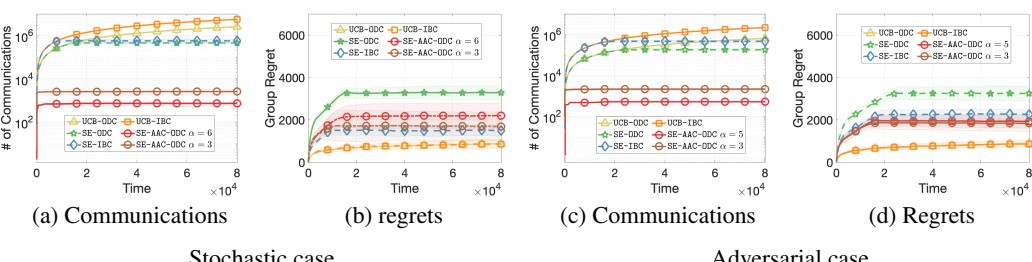

(a) Communications     (b) regrets     (c) Communications     (d) Regrets

Stochastic case        Adversarial case

Figure 6: Additional experiments for `SE-AAC-ODC` with different $\alpha$

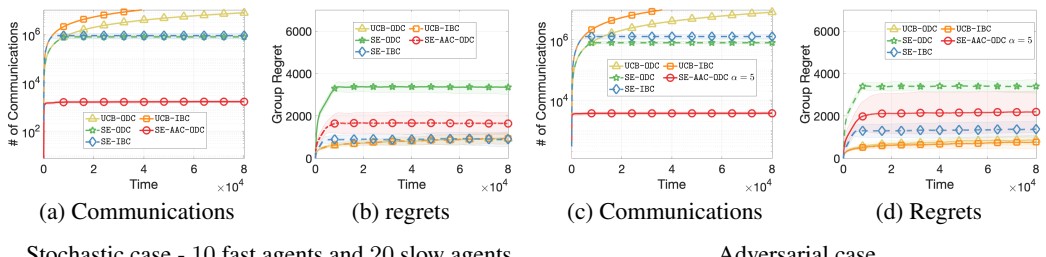

(a) Communications     (b) regrets     (c) Communications     (d) Regrets

Stochastic case - 10 fast agents and 20 slow agents        Adversarial case

Figure 7: Additional experiments for 30 agents

