# OpenReview forum: "Communication-Efficient Algorithm for Asynchronous Multi-Agent Bandits"
_ICLR.cc/2024/Conference — Submitted to ICLR 2024_

### Official Review · Reviewer_F9Jr · 2023-10-27

**Soundness:** 3 good
**Presentation:** 2 fair
**Contribution:** 3 good
**Rating:** 6
**Confidence:** 4

**Summary:**

In this paper, the authors study the cooperative asynchronous multi-agent multi-armed bandit problem:
- the agents are allowed to communicate for handling the same Bernoulli multi-armed problem,
- at each time step a subset of agents, which is chosen in advance by an adversary, can play the arms.
The authors designed a new communication protocol: an agent communicates if its new local observations enough reduce the estimation of the global confidence bound of the considered arm. The proposed bandit algorithm is based on Successive Elimination of suboptimal arms. The algorithm is analyzed and then tested versus the state-of-the-art.

**Strengths:**

The main claim of the paper is to handle cooperative asynchronous multi-agent multi-armed bandits with a constant communication cost, while guarantying a rate optimal regret upper bound.
This study is interesting, notably the fact that agents asynchronously play, which has a significant applicative potential. The constant communication cost can be easily obtained by an explore-then-exploit approach (Hillel et al 2013), but not with an optimal rate regret upper bound. So, it is an interesting result.

**Weaknesses:**

The paper has some weaknesses:
        - Objective function: the authors wrote that the expectation in the expected regret (actually, the pseudo regret) is taken over the randomness of the algorithm and reward realizations. However, the reviewer did not find where there is randomness in Algorithm 1. So, there is no expectation in the second line of the pseudo regret R(T).

        - Algorithm 1: the reviewer does not understand what lines 21-24 mean.
        - Communication cost: the reviewer is wondering why agents send an elimination message to other agents (line 7), while agents also send their estimates line 15.
        - Number of agents: in the experiments, the number of agents is lower than the number of arms.
        - Proof of Lemma 1: equation (a1), M_t is not defined and misleading.
        - Proof of Theorem 2, step 3: the reviewer does not understand where the last equation page 17 comes from. Could you provide more details?
        - Model: the authors assume that there is only on best arm (\mu_1 > \mu_2 \geq …). This is a strong assumption that limits the scope of this study.

The authors answered to most of my concerns. I raised my score to 6.

**Questions:**

See above

---

> ### Author Response · Authors · 2023-11-20
> **On Regret Definitioin**
>
> > Reviewer: Objective function: the authors wrote that the expectation in the expected regret (actually, the pseudo regret) is taken over the randomness of the algorithm and reward realizations. However, the reviewer did not find where there is randomness in Algorithm 1. So, there is no expectation in the second line of the pseudo regret R(T).
>
> **Answer:** The mention of randomness in the expected regret definition is a standard approach to cover a wide range of algorithms, including probabilistic ones. Although Algorithm 1 is deterministic, this inclusion allows our analysis to remain applicable to other potential algorithms, such as those based on stochastic methods like Thompson sampling based ones.

---

> ### Author Response · Authors · 2023-11-20
> **On Lines 21-24 in Algorithm**
>
> > Reviewer: Algorithm 1: the reviewer does not understand what lines 21-24 mean.
>
> **Answer:** Lines 21-22 are the receiving counterpart for the message sent in Line 16 (by other agents $m'$),
>       while Lines 23-24 are the receiving counterpart for the message sent in Line 15. Take the pair between Line 16 and Lines 21-22 as an example. Line 16 means that, when sending message from agent $m$ to agent $m'$, the agent $m$ sends the token $\texttt{tk}^{(m\to m')}$ to agent $m'$ as well.
>       Lines 21-22 means that, when receiving a token $\texttt{tk}^{(m'\to m)}$ that was used to send a message from agent $m'$ to agent $m$, agent $m$ keeps this token. This pair constitutes the token receiving and sending protocol. We will add this explanation to the final version of this paper.

---

> ### Author Response · Authors · 2023-11-20
> **On Communication Cost**
>
> > Reviewer: Communication cost: the reviewer is wondering why agents send an elimination message to other agents (line 7), while agents also send their estimates line 15.
>
> **Answer:** The seemly redundant communications are due to the asynchronous decision rounds of agents and the small number of communications (constant communications in total).
>       In synchronous cases, as all agents have the same number of decision rounds, sending reward mean estimates is enough.
>       However, when it comes to this paper's asynchronous case,
>       sending reward mean estimates (Line 15) is not enough.
>       Without the elimination message to other agents (Line 7),
>       some "fast" agents (high active frequency) may eliminate one arm much earlier than some "slow" agents: the number of reward observations of "fast" agents can be much larger than that of the "slow" ones, and the small number of communications achieved by our algorithm design can delay the observation sharing from "fast" agents to "slow" agents a lot. Therefore, the elimination message in Line 7 is important for reducing the regret cost due to "slow" agents (more general, due to asynchronicity) and thus is not redundant in an asynchronous setting.

---

> ### Author Response · Authors · 2023-11-20
> **On Number of Agents in Experiments**
>
> > Reviewer: Number of agents: in the experiments, the number of agents is lower than the number of arms.
>
> **Answer:** In Figure 7 in Appendix D of the revised manuscript, we report additional experiments where the number of agents is $M=30$ greater than the number of arms $K=16$. In these experiments, our algorithm outperforms other baseline algorithms in communication and enjoys similar regret as the other baselines, which is the same observation as in our original numerical simulations in Figure 1.

---

> ### Author Response · Authors · 2023-11-20
> **On $Mt$ in Proof of Lemma 1**
>
> > Reviewer: Proof of Lemma 1: equation (a1), M_t is not defined and misleading.
>
> **Answer:** In inequality (a1), the upper bound of summation is $M\times t$, not $M_t$, where $M$ is the number of agents and $t$ is the current time index. In the revised, we updated it to $M\cdot t$ to avoid misunderstanding.

---

> ### Author Response · Authors · 2023-11-20
> **On Step 3 of Proof of Theorem 3**
>
> > Reviewer: Proof of Theorem 2, step 3: the reviewer does not understand where the last equation page 17 comes from. Could you provide more details?
>
> **Answer:** The last inequality comes from the last equation, where we substitute $1=\texttt{ECR}\_t^{(m)}(0)$ and take $\log_\alpha$ on both sides.
>     Below, we illustrate how the last equation is derived, and we updated this proof detail in the revised version as well.
>     We note that the estimated confidence radius $\texttt{ECR}\_k^{(m)}(T)$ should be written as the $\texttt{ECR}_k^{(m)}(\tau_k(\kappa_k))$ at the last communication round $\tau_k(\kappa_k)$ because after the last communication, the width of $\texttt{ECR}$ has no further impact on communication.
>     \newline
>     Recall that $\tau_k(t)$ is the latest communication round about arm $k$ on or before time slot $t$.
>     Then, $\texttt{ECR}_k^{(m)}(\tau_k(\kappa_k))$ is the $\texttt{ECR}$ at the latest communication for arm $k$ which can be upper bounded as follows,
>     $$
>     \texttt{ECR}_k^{(m)}(\tau_k(\kappa_k))
>     \overset{(a)}=
>     \texttt{CR}(n_k(\tau_k(\kappa_k))) \ge
>     \texttt{CR}(n_k (\kappa_k))
>     \ge \frac{\Delta_k}{2(1+\alpha)},
>     $$
>     where equality (a) is because $\tau_k(\kappa_k)$ is a communication time slot in which the estimated confidence radius ($\texttt{ECR}$) is equal to $\texttt{CR}$.

---

> ### Author Response · Authors · 2023-11-20
> **On Single Optimal Arm Assumption**
>
> > Reviewer: Model: the authors assume that there is only on best arm (\mu_1 > \mu_2 \geq …). This is a strong assumption that limits the scope of this study.
>
> **Answer:** Assuming a single optimal arm is for the ease of presentation, which was widely assumed in literature (e.g., see Even-Dar et al. (2006, Section 2.1)). In the case of multiple optimal arms,
>       tailoring our current algorithm could also work.
>       In the original algorithm design, when only one arm remains in the candidate arm set (the optimal arm), the algorithm stops communication.
>       When it comes to the case of multiple optimal arms, the above communication stop condition is invalid.
>       Instead, one can set an exploration threshold, and when the number of times pulling the remaining arms in the candidate arm set exceeds this threshold, stop the communication. This threshold would guarantee that the total communications are still constant, as well as that the reward means of remaining arms are close enough to have a sublinear regret upper bound.
>
> ---
> Even-Dar, Eyal, et al. "Action Elimination and Stopping Conditions for the Multi-Armed Bandit and Reinforcement Learning Problems." Journal of Machine Learning Research 7.6 (2006).

---

> > ### Comment · Reviewer_F9Jr · 2023-11-22
> >
> > I thank you for your answers to my concerns.
> >
> > I still think that a single best arm is not a mild assumption. I am not sure that if \epsilon > 0, the optimal regret rate is preserved.
> >
> > In (Even-Dar et al 2006) Successive Elimination is presented with \epsilon=0. However, since this seminal paper they were a lot of papers handling (\epsilon,\delta) PAC algorithms for Best Arm Identification, including decentralized algorithms (see for instance the papers below).
> >
> >
> >
> > (Gabillon et al 2012) Victor Gabillon, Mohammad Ghavamzadeh, Alessandro Lazaric, Best Arm Identification: A Unified Approach to Fixed Budget and Fixed Confidence, NeurIPS 2012.
> >
> > (Kalyanakrishnan et al 2012) Shivaram Kalyanakrishnan, Ambuj Tewari, Peter Auer, Peter Stone, PAC Subset Selection in Stochastic Multi-armed Bandits, ICML 2012.
> >
> > (Kaufmann and Kalyanakrishnan 2013) Kaufmann and Kalyanakrishnan, Information Complexity in Bandit Subset Selection, ALT 2013.
> >
> > (Féraud et al 2020) Raphaël Féraud, Réda Alami, Romain Laroche, Decentralized Exploration in Multi-Armed Bandits, ICML 2020.

---

> > > ### Author Response · Authors · 2023-11-22
> > >
> > > We highly appreciate the reviewer for the follow-up suggestions. We acknowledge the need for a more detailed analysis regarding whether the threshold idea in [our reply](https://openreview.net/forum?id=ONnZVUrFBT&noteId=QXN3VP8391) could preserve the near-optimal regret.
> > > We emphasize that the paper assumes a single optimal arm for presentation simplicity, illustrating how to achieve constant communication with near-optimal regret in asynchronous multi-agent scenarios.
> > >
> > > As the reviewer suggests that these $(\epsilon, \delta)$-PAC algorithm designs after the seminal work (for a single optimal arm (Even-Dar et al 2006)) are feasible for multiple optimal arms, we believe that using these known techniques, together with our new algorithm design, would address the multiple optimal arms case, which is not the main focus of our paper, and the required additional analysis falls beyond our paper's scope.

---

### Official Review · Reviewer_andi · 2023-10-27

**Soundness:** 3 good
**Presentation:** 3 good
**Contribution:** 2 fair
**Rating:** 5
**Confidence:** 2

**Summary:**

This paper proposes a distributed asynchronous method for solving the multi-agent multi-armed bandit (MA2B) problem. The authors also extend their algorithms and results to preserve privacy. The main contribution of this paper is the design of the event-trigger-based asynchronous method to achieve higher communication efficiency. Both the theoretical and numerical results demonstrated the higher communication efficiency.

**Strengths:**

The paper is well written and easy to read. The idea of algorithm design is quite reasonable and sounds promising. They use the idea of "event trigger", namely, a node communicates only when it has enough new information to share. They also quantify the amount of information by something called "confidence radius". They also theoretically and numerically justify their main motivation: lower communication complexity. Their theoretical results show that their algorithm possesses the property of "constant communication", which is attractive to the reviewer.

**Weaknesses:**

1. Although the order of the regret bound of the proposed algorithm (see Remark 3) is the same as the optimal bound, the constant term before the order can be quite large. For example, when setting $\alpha=6$ as used in the experiments, the constant can be as large as 392, which is a very large number. I suggest the authors provide the regret bound (not only the order) of alternative methods and compare them. It will also be better to plot the regret bounds and the communication complexity bounds of the proposed method and alternative methods.

2. Can the authors clarify what is indeed the most important measurement (time or communication) in the MA2B task? The authors explained that synchronous updates will lead to redundant communication (Section 3.1). This is reasonable, but what is the final purpose of saving communication? Is it to achieve a lower regret within a shorter time horizon? If time is the most important thing, then by Figure 1,2, the performance of the proposed method is not as good as UCB-IBC.

3. I can see that from Theorem 2, a smaller alpha yields smaller regret. I can also see that from Figure 1,2, the proposed method has a higher regret compared to UCB-IBC. Therefore, I'm interested in the question of when we use smaller $\alpha$ to achieve similar regret as UCB-IBC, will the number of communications of the proposed method still be much smaller?

**Questions:**

I'm quite interested in the property of constant communication. By Theorem 2, the authors show that the regret bound is closely related to T, but the number of communications is independent of T. This means that communication between agents is not so important and even if there are only a few communications, smaller $R(T)/T$ can still be achieved by using a large T. Can the authors intuitively explain the philosophy behind this property?

I'm not familiar with this topic, but if the authors can clarify my concerns, then I'm willing to improve my score.

---

> ### Author Response · Authors · 2023-11-20
> **On Coefficient in Regret Upper Bound**
>
> > Reviewer: Although the order of the regret bound of the proposed algorithm (see Remark 3) is the same as the optimal bound, the constant term before the order can be quite large. For example, when setting $\alpha=6$
>       as used in the experiments, the constant can be as large as 392, which is a very large number. I suggest the authors provide the regret bound (not only the order) of alternative methods and compare them. It will also be better to plot the regret bounds and the communication complexity bounds of the proposed method and alternative methods.
>
> **Answer:** The leading term coefficients of $\texttt{SE-ODC}$ and SE-IBC in our experiment setting are $24$, which is better than the $392$ factor in our $\texttt{SE-AAC-ODC}$ algorithm. For one thing, we note that the larger constant coefficient of our regret upper bound comes from a more involved regret analysis for our constant communication result. This implies the challenge and novelty of our new algorithm.
>       For another thing, in our experiments, e.g., Figure 1(b), our algorithm with only constant communication has a better regret performance than that of $\texttt{SE-ODC}$ with logarithmic communication. That implies although there is a larger artificial factor in our regret upper bound result, the empirical regret performance of our algorithm is better (even with lower communications).

---

> ### Author Response · Authors · 2023-11-20
> **On Most Important Metric**
>
> > Reviewer: Can the authors clarify what is indeed the most important measurement (time or communication) in the MA2B task? The authors explained that synchronous updates will lead to redundant communication (Section 3.1). This is reasonable, but what is the final purpose of saving communication? Is it to achieve a lower regret within a shorter time horizon? If time is the most important thing, then by Figure 1,2, the performance of the proposed method is not as good as UCB-IBC.
>
> **Answer:** We acknowledge that this paper primarily delves into theoretical aspects, with the central focus on assessing the efficacy of communication cost reduction while maintaining near-optimal $O(K\log T)$ regret, referred to as "time" in the reviewer's question. It is essential to clarify that the primary goal is not to minimize regret further, as prior works, including our own, have already achieved near-optimal regret levels. While empirical results indicate that the proposed algorithm's regret may not match that of UCB-IBC, it is noteworthy that both UCB-IBC and our algorithm attain the same $O(K\log T)$ regret level.

---

> ### Author Response · Authors · 2023-11-20
> **On Impact of Parameter $\alpha$**
>
> > Reviewer: I can see that from Theorem 2, a smaller alpha yields smaller regret. I can also see that from Figure 1,2, the proposed method has a higher regret compared to UCB-IBC. Therefore, I'm interested in the question of when we use smaller
>  to achieve similar $\alpha$ regret as UCB-IBC, will the number of communications of the proposed method still be much smaller?
>
> **Answer:** We note that our algorithm only needs constant communications, which are far less than the linear communication cost of UCB-ICB; it is impossible for our algorithm to beat UCB-ICB in terms of regret metric.
>       For another thing, there is indeed a trade-off of regret and communication in our algorithm ($\texttt{SE-AAC-ODC}$) in terms of the parameter $\alpha$. In Appendix D of the revised manuscript, we compare the performance of $\texttt{SE-AAC-ODC}$ with $\alpha=3$ and $\alpha=6$ in Figure 6. This simulation validates the trade-off---the case of $\alpha=3$ enjoys a better regret but suffers a higher communication. Additionally, the communications of both cases are much lower than that of other baseline algorithms.

---

> ### Author Response · Authors · 2023-11-20
> **On Intuition of Constant Communication**
>
> > Reviewer: I'm quite interested in the property of constant communication. By Theorem 2, the authors show that the regret bound is closely related to T, but the number of communications is independent of T. This means that communication between agents is not so important and even if there are only a few communications, smaller
>       can still be achieved by using a large T. Can the authors intuitively explain the philosophy behind this property?
>
> **Answer:** The high-level intuition is that an agent only needs to know which arm is optimal to achieve sublinear regret in multi-armed bandits, and the communication cost of sharing this information (an arm index) is independent of the total number of decision rounds.
>       Therefore, if one fast agent (with a large number of active decision rounds) could somehow "smartly" find the optimal arm, then it only needs to pay a constant number of communications to disseminate the optimal arm index to all other agents.

---

> > ### Comment · Reviewer_andi · 2023-11-22
> > **Thank you for your reply**
> >
> > Dear authors,
> >
> > Thank you for your reply and I read all of them. Unfortunately, I decided to keep my score. I agree that the proposed algorithm has a lower communication complexity, but it is valuable only if it can achieve lower communication costs when achieving the same regret compared to alternative methods such as UCB-ICB.

---

> > > ### Author Response · Authors · 2023-11-22
> > >
> > > The authors respect the reviewer's assessment.
> > >
> > > It is impossible to devise an algorithm with lower communication (in asymptotic order) that could beat an IBC-based algorithm (assume the same decision-making policy) in terms of empirical regret performance. Recall that the immediate broadcast communication (IBC) policy shares all observations among all agents immediately, which is one extreme case of full communication. Therefore, no other communication policy (with lower communication cost) could achieve better empirical regret performance than IBC.
> > >
> > > Although our algorithm cannot outperform UCB-IBC in empirical regret performance, we emphasize that our algorithm and UCB-IBC both achieve the same near-optimal $O(K\log T)$ regret upper bound, and our algorithm only needs constant (time-independent) communications while UCB-IBC needs $O(T)$ communication.

---

### Official Review · Reviewer_Ttd7 · 2023-10-30

**Soundness:** 3 good
**Presentation:** 3 good
**Contribution:** 2 fair
**Rating:** 5
**Confidence:** 4

**Summary:**

This paper studies cooperative asynchronous MABs, such that the arm pulls are asynchronous across the agents. The authors proposed a fully distributed algorithm (called SE-AAC-ODC) which achieves near-optimal regret with the number of communications independent of the time horizon. Furthermore, the algorithms are modified to achieve local differential privacy along with rigorous guarantees. Numerical simulations are also presented to compare the performance of their algorithm with the other known baselines.

**Strengths:**

Originality
------------------------------------------------------------------------------------------------------------
- N/A
_______________________________________________________________________
Quality
--------------------------------------------------------------------------------------------------------------
- Analysis seems to be correct
__________________________________________________________________________
Clarity
---------------------------------------------------------------------------------------------------------------
- Algorithm and analysis are clearly explained
- The paper is well-written for the most part (minor concerns are mentioned in the Questions)
________________________________________________________________________________
Significance
---------------------------------------------------------------------------------------------------------------
- The multi-agent asynchronous MAB considered in this paper is motivated by real-world applications (covered in detail in the Introduction) and could be of interest to the researchers studying multi-agent bandits

**Weaknesses:**

- **Lack of originality:** The algorithm and the analysis are extensions of known works in (Yang et. al. 2023) and (Chen et. al. 2023)

- **Missing details in numerical simulations:** The algorithm AAE-ODC from (Chen et. al. 2023) uses a buffer threshold for messages communicated among the agents. There is no mention of the values of the buffer threshold used in AAE-ODC for the experiments presented in the paper
________________________________________________________________________________________
**Potentially misleading claim about the communication cost of the SE-AAC-ODC:**
---------------------------------------------------------------------------------------------------------------------------------------------------
The authors claim that the communication cost of SE-AAC-ODC, which scales as $O(KM\log \Delta^{-1})$ is much smaller than that of (Chen et. al. 2023), in which the communication cost scales as $O(KM^2 \Delta^{-2}\log T)$ as claimed in this paper. However, the authors haven't mentioned all the details from (Chen et. al. 2023). Recall from the previous bullet that the algorithms in (Chen et. al. 2023) use buffer thresholds.
- The communication cost scaling as $O(KM^2 \Delta^{-2}\log T)$ of the AAE-ODC algorithm in (Chen et. al. 2023) only holds when the buffer thresholds are constant with respect to the number of communications (ref: Corollary 1, part (b) in (Chen et. al. 2023)).
- However, if the buffer thresholds are exponential with respect to the number of communications, the communication cost of the AAE-ODC algorithm in (Chen et. al. 2023) scales as $O\big(M^2 \log \frac{K \log T}{\Delta^2}\big)$ (ref: Corollary 2, part (b) in (Chen et. al. 2023)).

I claim that when $K$ is very large (and $K > M$), the communication cost of the AAE-ODC algorithm in (Chen et. al. 2023) with the exponential buffer threshold is smaller than that of the communication cost of SE-AAC-ODC in this paper, unless the time horizon $T$ scales doubly exponentially in $K$. It can be quickly noticed by taking the ratio of the communication costs $\frac{M^2 \log \frac{K \log T}{\Delta^2}}{KM\log \frac{1}{\Delta}} = \frac{M \log \frac{K \log T}{\Delta^2}}{K\log \frac{1}{\Delta}}$, which is less than some small constant (ignoring other constants in the $O$ notation) for $T=O(\exp (K^{-1}\exp(K)))$ (assuming natural logarithm). This makes the constant communication cost of SE-AAC-ODC algorithm in this paper vacuous.

- The preceding discussion also puts the validity of the numerical experiments into question.

**Questions:**

- I encourage the authors to address the concerns in the Weaknesses section, in particular about the numerical results and misleading claim about the communication cost.
- Can the authors also provide a lower bound on the communication cost of the asynchronous multi-agent MAB model considered in this paper?

Minor Comments
----------------------------------------------------------------------------------------------------------------------
- In the Introduction, the authors have used $\mathbb{N}^{+}$ for the set of natural numbers. The $+$ in the superscript is redundant, as natural numbers are positive by definition.
- In the Related Work (Section 1.2), (Chawla et. al. 2020) considers gossip-style communication scheme as well, which isn't mentioned.
- In the definition of the confidence radius in eq (1), Section 3.2.1, shouldn't it be $\min (1, \sqrt{\frac{2 \log T}{n}})$?
- In Remark 10, the bound $\frac{e^{\epsilon}+1}{e^{\epsilon}-1} \leq 1 + \frac{1}{\epsilon}$ is incorrect. To check this, set $\epsilon = 1$ and notice that the left hand side $> 2$ and the right hand side $= 2$. The correct bound is $\frac{e^{\epsilon}+1}{e^{\epsilon}-1} \leq 1 + \frac{2}{\epsilon}$. It can be proved by noticing that $\frac{e^{\epsilon}+1}{e^{\epsilon}-1} - 1 = \frac{2}{e^{\epsilon}-1} \leq \frac{2}{\epsilon}$, since $e^{\epsilon}-1 \geq \epsilon$.
________________________________________________________________________________________________________
After detailed discussions with the authors, I have increased my score from 3 to 5.

---

> ### Author Response · Authors · 2023-11-20
> **On Novelty**
>
> > Reviewer: Lack of originality: The algorithm and the analysis are extensions of known works in (Yang et. al. 2023) and (Chen et. al. 2023)
>
> **Answer:** We acknowledge the foundational role of Yang et al. (2023) and Chen et al. (2023) in our algorithmic design. However, our analysis significantly extends their work, addressing challenges unique to our setting.
>       On the one hand, the analysis of Chen et al. (2023) cannot be used in our case, as we are aiming for a constant communication upper bound, which is intrinsically different from their time-dependent bounds.
>       On the other hand, the analysis tool in Yang et al. (2023) relies heavily on the synchronous multi-agent setting (agents always have the same number of samples), which is invalid in our asynchronous setting. In this paper, we develop a new analysis approach that relies on the ''fastest'' agent (with the highest number of active decision rounds, see Lemma 1).
>       We refer the reviewer to Section 1.2 (related works) for a more comprehensive discussion about how this work differs from (Yang et al. 2023) and (Chen et al. 2023).

---

> ### Author Response · Authors · 2023-11-20
> **On Details in Numerical Simulations**
>
> > Reviewer: Missing details in numerical simulations: The algorithm AAE-ODC from (Chen et. al. 2023) uses a buffer threshold for messages communicated among the agents. There is no mention of the values of the buffer threshold used in AAE-ODC for the experiments presented in the paper
>
> **Answer:** For the experiments in the main paper, we set the buffer of $\texttt{SE-ODC}$ ($\texttt{AAE-ODC}$) as constant $1$ (that is, immediately communicate when there is a demand without any buffer). In Figure 5 of Appendix D, we report the $\texttt{SE-ODC}$ with no buffer (green), and $\texttt{SE-ODC-D}$ with doubling buffer (brown) and compare both to our algorithm, $\texttt{SE-AAC-ODC}$ (red).
>       Although $\texttt{SE-ODC-D}$ with doubling buffer enjoys better communication performance than $\texttt{SE-ODC}$ with constant buffer, $\texttt{SE-ODC-D}$'s communication performance is still worse than that of $\texttt{SE-AAC-ODC}$ proposed in our paper, and $\texttt{SE-ODC-D}$'s empirical regret is far worse than $\texttt{SE-AAC-ODC}$, which is because $\texttt{SE-ODC-D}$'s regret upper bound $O(KM\log T)$ is intrinsically worse than our algorithm's $O(K\log T)$ regret upper bound.
>       We updated these additional simulation results in the revised version of this paper.

---

> ### Author Response · Authors · 2023-11-20
> **On Comparing Communication Cost of SE-AAC-ODC to (Chen et al., 2023)**
>
> > Reviewer: Potentially misleading claim about the communication cost of the SE-AAC-ODC:
>
> **Answer:** It is unfair to compare the result in Corollary 2(b) (Chen et al., 2023) with our algorithm.
>       This is because the result in Corollary 2(b) (Chen et al., 2023) only holds when the buffer thresholds are exponential  (e.g., doubling in $\texttt{SE-ODC-D}$ in the above simulation).
>       But if the buffer thresholds are exponential, the algorithm in Chen et al. (2023) suffers a $O(KM\log T)$ regret which is away from the near-optimal regret by a factor of the number of agents $M$ (see their regret upper bound's second term in Theorem 1 (b)).
>       In fact, this $O(KM\log T)$ regret upper bound can be achieved by all agents individually running UCB *without any communication*.
>       Hence, it is unfair to compare the result in Corollary 2(b) (Chen et al., 2023) with our algorithm that always achieves the near-optimal $O(K\log T)$ regret.

---

> > ### Comment · Reviewer_Ttd7 · 2023-11-20
> >
> > I thank the authors for the detailed justifications to my concerns and questions. I saw the second term in Theorem 1(b) in (Chen et. al. 2023), and I am still not convinced with the response provided by the authors. This is because from Theorem 1(b) in (Chen et. al. 2023), the scaling of $G_{i}^{j}$ in the second term isn't clear when the buffer thresholds are exponential with respect to the number of communications. Even though the empirical results provided by the authors for the doubling buffer in AAE-ODC support their assertion, this is a theory paper. So, I would like to see whether $G_{i}^{j} = \frac{16 \alpha \log T}{\Delta_{i}^{2}}$ for any choice of exponent while using the exponential buffer in Theorem 1(b) in (Chen et. al. 2023). This is because if there exists some choice of exponent for the exponential buffer such that $G_{i}^{j}$ is a constant dependent only on system parameters and independent of time $T$, then my assessment will stand as is.

---

> > > ### Author Response · Authors · 2023-11-20
> > > **Validate $G_i^j= \frac{8\alpha\log T}{\Delta_i^2}$ when buffer is exponential**
> > >
> > > We thank the reviewer for their timely response and appreciate their participation in the discussion. Below, we show that, if the buffer increases in an exponential manner (i.e., the case in Corollary 2(b) of Chen et al. (2023)), in general, one would have $G_i^j= \frac{8\alpha\log T}{\Delta_i^2}$, and, therefore, the algorithm in Chen et al. (2023) would suffer a suboptimal $O(KM\log T)$ regret upper bound.
> > >
> > > Recall that
> > > $$G\_i^j=\min\left\\{\sum\_{j'\in\mathcal{A}\setminus\{j\}}f(c\_{\tau\_{i}}\^{j'\to j}), \frac{8\alpha\log T}{\Delta\_i^2}\right\\}$$
> > > By the definition of $\tau_i$ in Chen et al. (2023), we know that, in the worse case, $$c_{\tau_{i}^{j'\to j}}\le \frac{2\alpha\log T}{\Delta_i^2}.$$
> > > When the buffer size is exponential, i.e., $f(c)=a^{c-1}$ where $a$ is the exponent (e.g. if it is doubling, $a=2$). Then, we have $$f(c_{\tau_{i}}^{j'\to j})\le T a^{\frac{2\alpha}{\Delta_{i}^{2}\log_{a}e}},$$
> > > where $e$ is the nature constant and the base for $\log T$. Therefore, we have
> > > $$G\_i^j=\min\left\\{a\^{\frac{2\alpha}{\Delta\_{i}^{2}\log\_{a}e}}\cdot\sum\_{j'\in\mathcal{A}\setminus\{j\}}T, \quad\frac{8\alpha\log T}{\Delta_i^2}\right\\}.$$
> > > That means, when the total number of decision rounds $T$ is large enough, we have $$G_i^j= \frac{8\alpha\log T}{\Delta_i^2}.$$

---

> > > > ### Comment · Reviewer_Ttd7 · 2023-11-21
> > > > **Follow up**
> > > >
> > > > 1. Minor comment: isn't $G_{i}^{j} = \min \\{\sum_{j' \in \mathcal{A} \backslash j} f(c_{\tau_{i}}^{j' \rightarrow j}), \frac{16 \alpha \log T}{\Delta_{i}^{2}}\\}$? The $8$ seems to be a typo.
> > > > 2. How do you bound $c_{\tau_{i}}^{j' \rightarrow j}$ in the second step? Is it tight?
> > > > 3. What is the value of $M$ (# of agents) in Figure 5 (SE-ODC with doubling buffer) in the appendix, section D?

---

> > > > > ### Author Response · Authors · 2023-11-21
> > > > >
> > > > > 1. It is $8\alpha$ in Chen et al. (2023, Eq. (5)).
> > > > > 2. This bound is due to the definition of $\tau_i$ in the first paragraph of Chen et al. (2023, Section 4). This is tight in terms of the number of decision rounds $T$. One can consider a case that two of $M$ agents are active every other time slot, where the on-demand communications $c_{\tau_i}^{(j'\to j)}$ are $O(\log T)$ for these two agents.
> > > > > 3. $M=16$ in Figure 5. We note that although our algorithm removed an $M$ factor in the regret upper bound of Chen et al. (2023, Corollary 2(b)), our algorithm's regret upper bound has a slightly larger constant coefficient due to lower constant communication. This may explain why one cannot observe the $M$ times larger regert of Chen et al. (2023, Corollary 2(b)) than ours in Figure 5. See [this reply](https://openreview.net/forum?id=ONnZVUrFBT&noteId=vLt3ZK3tPe) for more details about the coefficient.

---

> > > > > > ### Comment · Reviewer_Ttd7 · 2023-11-21
> > > > > >
> > > > > > Thank you for the clarification. From Theorem 2, it is clear that there is a regret communication trade-off, dictated by the threshold parameter $\alpha$ (when $\alpha$ is large, less communication but large group regret and vice-versa when $\alpha$ is small). I agree that the trade-off between communication and group regret is unavoidable. However, the communication parameter $\alpha$ affects the leading order $\log T$ term in group regret. For example, if $\alpha = O(\log T)$, the group regret of the proposed algorithm scales as $O((\log T)^3)$, which is bad and the regret optimality (mentioned in Remark 3) does not hold anymore. Based on the results, it raises the question of whether we can come up with another algorithm which achieves constant communication while impacting only the time independent terms (or "constant" as used by the authors) in group regret. If that is not achievable, I am back to my question of lower bound on the number of communications. Hence, I keep my score.

---

> > > > > > > ### Author Response · Authors · 2023-11-21
> > > > > > > **Clarification for a misunderstanding**
> > > > > > >
> > > > > > > > Reviewer: Based on the results, it raises the question of whether we can come up with another algorithm which achieves constant communication while impacting only the time independent terms (or "constant" as used by the authors) in group regret.
> > > > > > >
> > > > > > > **Answer:** This is exactly what our paper ($\texttt{SE-AAC-ODC}$) achieves! For example, if $\alpha = 2$ in Theorem 2 of our paper, then our regret is near-optimal as follows,
> > > > > > > $$\sum_{k>1}\frac{72\log T}{\Delta_k} + \sum_{k>1} M\Delta_k + KM^2$$
> > > > > > > , and our communication, upper bound by
> > > > > > > $$\sum_{k>1}2M\log_2 \left( \frac{6}{\Delta_k} \right) + 2M\log_2 \left( \frac{6}{\Delta_2} \right)
> > > > > > >         + 2KM^3,$$
> > > > > > > is independent of time.
> > > > > > >
> > > > > > > We also refer to the reviewer to [our reply](https://openreview.net/forum?id=ONnZVUrFBT&noteId=h103ugKyyz) about the communication lower bound discussion.

---

> > > > > > > > ### Comment · Reviewer_Ttd7 · 2023-11-21
> > > > > > > >
> > > > > > > > The SE-AAC-ODC algorithm does not achieve near optimal regret for all values of $\alpha$. I refer the authors to the case of $\alpha = \log T$ (I can choose any value for $\alpha$ as long as $\alpha > 1$, right?) which I mentioned in my previous comment.

---

> > > > > > > > > ### Author Response · Authors · 2023-11-21
> > > > > > > > >
> > > > > > > > > The parameter $\alpha$ is an algorithm input parameter decided by the algorithm designer. As we are aiming for good theoretical results, it would be better to choose a good $\alpha$, e.g., $\alpha =2$, instead of adversarial choosing the input.
> > > > > > > > >
> > > > > > > > > For example, in the well-known (single agent) UCB algorithm, the UCB index has a constant parameter before its confidence radius term, which is usually to be set as a constant in order to make UCB achieve near-optimal regret. If one adversarially sets the constant parameter as $O(\log T)$ before the confidence radius term in UCB, then UCB would also have a suboptimal regret upper bound.

---

> ### Author Response · Authors · 2023-11-20
> **On Communication Lower Bound**
>
> > Reviewer: Can the authors also provide a lower bound on the communication cost of the asynchronous multi-agent MAB model considered in this paper?
>
> **Answer:**
>       Providing a non-trivial lower bound for communication in asynchronous multi-agent MAB settings is highly challenging and remains an open problem in the literature.
>       This is because some extremely asynchronous scenarios could make communication unnecessary. For example, consider the case that one agent is active in all time slots, while all other agents are only active in one single time slot.
>       In this case, there is no need to conduct communication among these agents, that is, one can achieve near-optimal regret by all agents individually running UCB without any communication.
>
> On the other hand, the only related communication lower bound results that we are aware of is in Wang et al. (2020, Section 3.4), where they show that to achieve a $o(M\sqrt{KT})$ regret upper bound in *synchronous* multi-agent MAB, the expected communication is at least $\Omega(M)$.
>       If we exclude the extreme cases and only consider mildly asynchronous scenarios, then the $\Omega(M)$ lower bound
>       could be used to understand the tightness of our communication upper bound.
>       That implies that
>       our communication upper bound $O(KM\log(\Delta^{-1}))$ is tight in terms of the number of agents $M$.
>
> In a word, proving a non-trivial communication lower bound is challenging and beyond the scope of this work. Comparing with known results suggests that our communication upper bound is tight in terms of $M$.
>
>
>
> ---
> - Yuanhao Wang, Jiachen Hu, Xiaoyu Chen, and Liwei Wang. Distributed bandit learning: Nearoptimal regret with efficient communication. In 8th International Conference on Learning Representations, ICLR 2020, Addis Ababa, Ethiopia, April 26-30, 2020, 2020b.

---

> ### Author Response · Authors · 2023-11-20
> **On Minor Comments**
>
> We highly appreciate the reviewer for suggesting other minor changes. All were addressed in the revised version, except for $\mathbb{N}^+$. Because whether the natural number contains zero is ambiguous (see the first sentence of the nature number on the Wikipedia page), the authors think it would be better to use $\mathbb{N}^+$ to avoid misunderstanding.

---

> ### Comment · Reviewer_Ttd7 · 2023-11-21
>
> Fair enough, although the dependence of communication parameter $\alpha$ on the leading order term $\log T$ in group regret (as $(1 + \alpha)^2$) isn't ideal in my opinion. I am raising my score to 5, but can't vouch for acceptance of the paper because of the aforementioned concern.

---

> > ### Author Response · Authors · 2023-11-21
> >
> > Thank you very much. Truly appreciate your active participation in the discussion.

---

### Official Review · Reviewer_vstV · 2023-10-30

**Soundness:** 2 fair
**Presentation:** 3 good
**Contribution:** 3 good
**Rating:** 6
**Confidence:** 3

**Summary:**

In this paper, the authors study the cooperative asynchronous multi-agent multi-armed bandits problem, where the active agents in each round are unknown in advance. The authors propose a new algorithm with Accuracy Adaptive Communication (AAC) protocol that achieves near-optimal regret and requires communication rounds that are independent of time. The proposed approach is shown to be superior in terms of communication rounds through synthetic data.

**Strengths:**

* The paper is well-written, easy to follow, and well-organized.
* Compared to previous work [1], the authors propose a novel and superior Successive Elimination (SE) algorithm with Accuracy Adaptive Communication (AAC) protocol. The proposed algorithm achieves constant communication rounds, which are independent of time complexity. This is a significant advantage for real-world applications.
* The proposed algorithm is easy to implement and has the potential to be applied to a wide range of real-world applications.
* Synthetic data experiments show that the proposed algorithm outperforms existing algorithms in terms of communication rounds.

[1] Yu-Zhen Janice Chen, Lin Yang, Xuchuang Wang, Xutong Liu, Mohammad Hajiesmaili, John C.S.
Lui, and Don Towsley. On-demand communication for asynchronous multi-agent bandits. In
International Conference on Artificial Intelligence and Statistics, pp. 3903–3930. PMLR, 2023.

**Weaknesses:**

* The proposed algorithm is not validated with real-world data.
* The authors do not provide a lower bound for the number of communication rounds required, which makes it difficult to assess the performance of the proposed algorithm.

**Questions:**

Typos:

1.1 Success Elimination -> Successive Elimination

---

> ### Author Response · Authors · 2023-11-20
> **On Real-World Data in Simulation**
>
> > Reviewer: The proposed algorithm is not validated with real-world data.
>
> **Answer:** In numerical experiments, we set arm reward means as the click-through-rate from an ad-click Kaggle dataset (Avito Context Ad Clicks, 2015. https://www.kaggle.com/c/avito-context-ad-clicks), which simulates the real click-through process in online advertising applications. We updated this in the revised manuscript.

---

> ### Author Response · Authors · 2023-11-20
> **On Communication Lower Bound**
>
> > Reviewer: The authors do not provide a lower bound for the number of communication rounds required, which makes it difficult to assess the performance of the proposed algorithm.
>
> **Answer:**
>       Providing a non-trivial lower bound for communication in asynchronous multi-agent MAB settings is highly challenging and remains an open problem in the literature.
>       This is because some extremely asynchronous scenarios could make communication unnecessary. For example, consider the case that one agent is active in all time slots, while all other agents are only active in one single time slot.
>       In this case, there is no need to conduct communication among these agents, that is, one can achieve near-optimal regret by all agents individually running UCB without any communication.
>
> On the other hand, the only related communication lower bound results that we are aware of is in Wang et al. (2020, Section 3.4), where they show that to achieve a $o(M\sqrt{KT})$ regret upper bound in *synchronous* multi-agent MAB, the expected communication is at least $\Omega(M)$.
>       If we exclude the extreme cases and only consider mildly asynchronous scenarios, then the $\Omega(M)$ lower bound
>       could be used to understand the tightness of our communication upper bound.
>       That implies that
>       our communication upper bound $O(KM\log(\Delta^{-1}))$ is tight in terms of the number of agents $M$.
>
> In a word, proving a non-trivial communication lower bound is challenging and beyond the scope of this work. Comparing with known results suggests that our communication upper bound is tight in terms of $M$.
>
>
>
> ---
> - Yuanhao Wang, Jiachen Hu, Xiaoyu Chen, and Liwei Wang. Distributed bandit learning: Nearoptimal regret with efficient communication. In 8th International Conference on Learning Representations, ICLR 2020, Addis Ababa, Ethiopia, April 26-30, 2020, 2020b.

---

### Meta-Review · Area_Chair_M2xG · 2023-12-12

**Metareview:**

This paper focuses on communication between agents in cooperative asynchronous multi-agent multi-armed bandits.

This is an interesting and very active field, but the contributions of this paper are slightly incremental with respect to the existing literature. This is not a criticism on the quality and validity of the results (even though a reviewer pointed out during the discussion that some additional term in O(K / \Delta \log (KT/\Delta)) due to the use of successive elimination should appear... The authors should check this out).

This was therefore a borderline paper, and after discussion, the reviewers and I deemed that it does not reach the high ICLR bar.

**Justification For Why Not Higher Score:**

N/A

**Justification For Why Not Lower Score:**

N/A

---

### Decision · Program_Chairs · 2024-01-16

Reject